# Omni-Perception Policy Optimization for Multimodal Emotion Reasoning

Zhiyuan Han [1 2 3]   Beier Zhu [1]   Wenwen Tong [2]   Pengyang Shao [4]   Peipei Song [1]   Xinyi Wang [1]   Jiangnan Chen [2]
Lewei Lu [2]   Xun Yang [1]

## Abstract

We find that current emotion-oriented Omni-MLLMs still lack *reliable omni-modal perception*: they (i) underutilize multimodal cues in their reasoning trajectories and (ii) exhibit unfaithful behavior, often hallucinating modality-specific statements from other modalities. Building on these insights, we propose OPPO (Omni-Perception Policy Optimization), a reinforcement learning framework that explicitly optimizes multimodal perception. First, an Omni-Perception Reward decomposes ground-truth reasoning into fine-grained visual, acoustic, and emotion cues and rewards trajectories that semantically recover these cues. Second, an Omni-Perception Loss compares the policy under full and unimodally masked inputs, applying a KL penalty only to modality-specific evidence tokens to suppress cross-modal hallucination. We further introduce MEP-Bench, a diagnostic benchmark that quantifies *utilization* and *faithfulness*. Experiments show that OPPO achieves state-of-the-art performance on MER-UniBench and MME-Emotion, while substantially improving utilization and faithfulness scores on MEP-Bench, highlighting the importance of sufficient and faithful omni perception for multimodal emotion reasoning.

## 1. Introduction

Recent advances in Omni-MLLMs such as GPT-4o (Hurst et al., 2024) and Qwen-Omni (Xu et al., 2025a;b) are driving a paradigm shift in multimodal emotion recognition. Traditionally, affective computing (Lian et al., 2023a; 2026) has mainly focused on predicting emotion labels from multimodal signals. In contrast, by seamlessly integrating visual, acoustic, and linguistic streams, Omni-MLLMs enable *Multimodal Emotion Reasoning* (MER) (Lian et al., 2023b; 2025a), where models produce emotion predictions together with explicit reasoning grounded in multimodal cues. This shift not only improves the interpretability of emotion decisions, but also enhances prediction reliability via step-by-step deliberation.

While pure text reasoning has been extensively reinforced by methods such as GRPO and PAPO (Guo et al., 2025; Yu et al., 2025), MER further requires **reliable perception** of visual and acoustic signals—an ability that remains largely underexplored. We argue that reliable omni-modal perception in MER hinges on two principles: **(1) Utilization**: the reasoning trajectory should sufficiently leverage visual and acoustic cues. For example, a model that predicts *angry* just because the voice is loud, while ignoring a neutral facial expression and non-aggressive speech, violates this principle. **(2) Faithfulness**: modality-specific statements in the reasoning should be causally attributable to, and directly verifiable from, their source modality, rather than extrapolated from other modalities or spurious audio–video correlations. For instance, a model that asserts "the video shows a frown" when the face is actually neutral, simply because the voice sounds sad, violates this principle, as the visual claim is hallucinated from audio context.

To investigate how well existing emotion Omni-MLLMs satisfy these two principles, we construct the Multimodal Emotion Perception Benchmark (dubbed as MEP-Bench). Specifically, to quantify *utilization*, MEP-Bench computes the recall of human-annotated multimodal cues within the generated reasoning trajectories. To quantify *faithfulness*, MEP-Bench adopts a POPE-style evaluation (Li et al., 2023) under unimodal masking (*e.g.*, masking the video and asking, "Does the video show a frown?"). This evaluates whether visual statements are causally attributable to the video input rather than inferred from audio context or spurious audio–video correlations.

In Figure 1, we present both qualitative and quantitative results on MEP-Bench, showing that existing emotion Omni-MLLMs often *violate both principles*. For *utilization*, Figure 1 (a) shows that the video and audio jointly convey

---

[1]University of Science and Technology of China [2]SenseTime Research [3]Institute of Artificial Intelligence, Hefei Comprehensive National Science Center [4]National University of Singapore. Correspondence to: Xun Yang <xyang21@ustc.edu.cn>, Beier Zhu <beier.zhu@ustc.edu.cn>.

*Proceedings of the 43rd International Conference on Machine Learning*, Seoul, South Korea. PMLR 306, 2026. Copyright 2026 by the author(s).

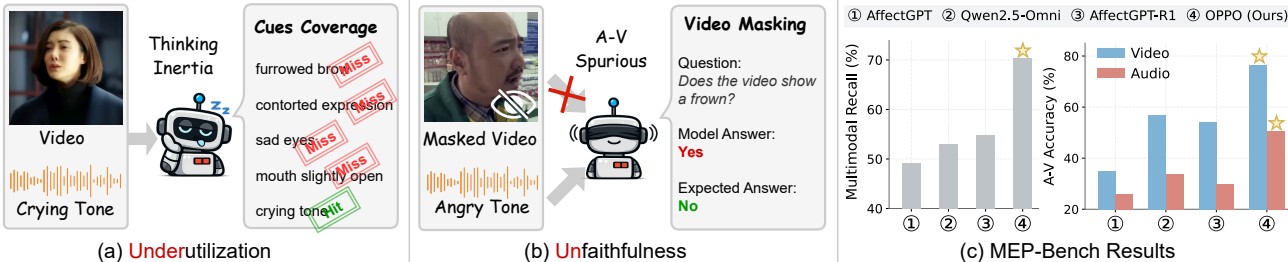

*Figure 1.* **Existing emotion Omni-MLLMs lack reliable perception of visual and acoustic cues.** (a) Underutilization (violating the *Utilization* principle). An empirical case from `MEP-Bench` where AffectGPT-R1 (Lian et al., 2025c) overlooks fine-grained visual cues (*e.g.*, furrowed brow, sad eyes) and mainly relies on the crying tone in the audio due to "thinking inertia". (b) Unfaithfulness (violating the *Faithfulness* principle). An empirical case where AffectGPT-R1 hallucinates a "frown" in the video based on an angry tone and spurious audio–visual correlations, even when the video is masked. (c) Left: multimodal recall on `MEP-Bench`; baseline models capture only about half of the human-annotated cues, whereas our `OPPO` achieves substantially higher recall. Right: accuracy of unimodal-masking probes on `MEP-Bench`; our `OPPO` attains much higher faithfulness on both video and audio masked questions.

multiple emotion-related cues, yet the baseline (Lian et al., 2025c) only picks up the "crying tone" in its reasoning chain. This under-utilization problem is further confirmed by the recall of multimodal cues (Figure 1 (c)), where current models achieve only around 50%, showing that they miss roughly half of the human-perceived cues; For *faithfulness*, Figure 1 (b) shows that, even when the video is masked and we ask "Does the video show a frown?", the baseline still answers "*Yes*" by appealing to the angry tone in the audio. Quantitatively, Figure 1 (c) shows that such probes yield only 35.0–54.2% accuracy on video questions and 26.0–30.0% on audio questions for existing models.

To bridge this gap, we propose Emotional `Omni-Perception Policy Optimization` (`OPPO`), a reinforcement learning framework (RL) that enforces grounded reasoning. First, to encourage better *utilization* of multimodal cues, we introduce **Omni-Perception Reward**. Specifically, we decompose the ground-truth reasoning into fine-grained visual, acoustic, and emotion cues, and segment the model's generated thinking into a sequence of clauses. We embed both cues and clauses and compute semantic similarities to determine whether each cue is covered by at least one clause. The Omni-Perception Reward then increases with the coverage of visual, audio, and emotion cues, encouraging the model to explicitly recover human-annotated evidence in its reasoning. Second, to enforce better *faithfulness*, we introduce **Omni-Perception Loss**. Specifically, we construct unimodally masked counterfactual inputs (*e.g.*, partially removing the video or audio) and compare the policy's behavior under the full and masked inputs. Unlike prior work such as PAPO (Wang et al., 2026c), which encourages all tokens to change under masking, we use cue matching to identify tokens that describe visual or acoustic evidence and apply a KL-based penalty only to these modality-specific tokens, encouraging their distributions to change substantially when the corresponding modality is masked. This forces statements about a modality to truly depend on its input and helps suppress cross-modal hallucination.

Extensive experiments show that `OPPO` achieves SoTA performance on the comprehensive MER-UniBench (Lian et al., 2025a) and MME-Emotion (Zhang et al., 2026a), while yielding consistent gains on the `MEP-Bench`. These results highlight the importance of sufficient and faithful omni perception for multimodal emotion reasoning. Our contributions are three-fold:

- **Findings:** We construct `MEP-Bench` and reveal that existing emotion Omni-MLLMs (1) underutilize multimodal cues in their reasoning trajectories and (2) exhibit unfaithful behavior under unimodal masking, often due to spurious cross-modal correlations.

- **Methodology:** We propose `OPPO`, an RL framework that combines an explicit Omni-Perception Reward for cue coverage with a mask-based Omni-Perception Loss applied to modality-specific evidence tokens, encouraging cue-grounded reasoning and suppressing hallucinated modality evidence.

- **Performance:** We achieve SoTA performance on multiple emotion benchmarks and substantially improve perception-oriented diagnostics, validating the effectiveness of `OPPO` for multimodal emotion reasoning.

## 2. Related Work

**Multimodal emotion reasoning.** Recent multimodal large language models (Wang et al., 2026b; 2025) have shown growing potential for emotion-related understanding (Xu et al., 2026; Dai et al., 2025; Zhang et al., 2026b; Song et al., 2024; Ye et al., 2025), driving multimodal emotion recognition to shift from closed-set label prediction (Lian et al., 2023a; Jiang et al., 2020; Liu et al., 2022a) toward open-vocabulary Multimodal Emotion Reasoning (MER) (Lian et al., 2023b; 2025a). Early MER efforts (Lian et al., 2023b; Cheng et al., 2024; Han et al., 2025; Lian et al., 2025a) primarily treated explanation generation and emotion pre-

diction as a multi-task learning problem based on supervised fine-tuning. Inspired by the DeepSeek-R1 paradigm (Guo et al., 2025), recent advances (Lian et al., 2025c; Yang et al., 2025; Zhao et al., 2025a) have unified these tasks by treating reasoning as an explicit Chain-of-Thought (CoT) process. Despite this unification, these models still lack *reliable omni-modal perception*. Notably, AffectGPT-R1 (Lian et al., 2025c) found that naively incorporating perception-related rewards could even degrade final prediction accuracy. This highlights `OPPO` as an essential solution, demonstrating that sufficient and faithful omni-perception is the key to unlocking robust reasoning and superior performance in multimodal emotion tasks.

**Reinforcement learning for MLLMs.** Reinforcement learning (RL) has been widely adopted to enhance the reasoning capabilities of MLLMs (Guo et al., 2025; Yu et al., 2025; Liu et al., 2025; Zhao et al., 2025b; Wang et al., 2026a; Zhao et al., 2026). However, ensuring reliable perception within reasoning trajectories remains an open challenge. Current frameworks have explored LLM-as-a-judge for quality assessment, such as HumanOmni-V2 (Yang et al., 2025), AffectGPT-R1 (Lian et al., 2025c), and Perception-R1 (Xiao et al., 2025), or employed unimodal masking with global penalties as in PAPO (Wang et al., 2026c). These approaches face inherent limitations: coarse-grained rewards often fail to ensure the utilization of specific multimodal cues, while the indiscriminate global penalties in PAPO can destabilize optimization and allow models to bypass genuine grounding. To address these technical gaps, we propose `OPPO`, which explicitly optimizes for two core principles: *utilization* and *faithfulness*. By integrating a fine-grained Omni-Perception Reward for exhaustive cue coverage with a targeted Omni-Perception Loss exclusively for modality-specific tokens, `OPPO` enforces the precise alignment between perceived evidence and reasoning, leading to more reliable multimodal reasoning.

## 3. Method

We formulate Multimodal Emotion Reasoning (MER) as an open-vocabulary generation task. Given a multimodal input $X = \{V, A, T\}$, where $V$, $A$, and $T$ represent the visual, acoustic, and textual streams, respectively, the model $\pi_\theta$ generates a structured sequence $Y$. To explicitly capture the reasoning process, the output is constrained to include a reasoning chain wrapped in `<think>...</think>` tags, followed by the final prediction enclosed in `<answer>...</answer>` tags.

Our pipeline starts with standard supervised fine-tuning (SFT) and is followed by a GRPO-style (Guo et al., 2025) reinforcement learning (RL) phase. We randomly select 5k samples from the MER-Caption+ (Lian et al., 2025a) for SFT and use the remaining samples for RL. During RL, we

follow Lian et al. (2025a), adopting the same emotion-wheel metric (details in Appendix C.1) to compute the F1 score as the task reward $R_{\text{acc}}$, together with a format reward $R_{\text{fmt}}$ to ensure structurally valid outputs (details in Appendix C.2).

To address the challenges of underutilization and unfaithfulness in multimodal perception, we augment the RL stage with `Omni-Perception Policy Optimization` (`OPPO`). Our `OPPO` introduces an explicit Omni-Perception Reward ( Section 3.1) that encourages better utilization of multimodal cues, together with an Omni-Perception Loss ( Section 3.2) that promotes faithful unimodal perception and discourages cross-modal hallucination.

### 3.1. Omni-Perception Reward

To encourage better utilization of multimodal cues in the reasoning process, we design an Omni-Perception Reward that measures how well the model's intermediate thoughts recover the ground-truth evidence.

**Multimodal evidence extraction.** For each training sample from the MER-Caption+ dataset (Lian et al., 2025a), we derive a set of atomic evidence. Specifically, we leverage GPT-5 (Singh et al., 2025) to extract fine-grained evidentiary statements explicitly present in each ground-truth reasoning paragraph, without introducing new evidence. Each statement is then categorized into one of three types, forming $\mathcal{C} = \{\mathcal{C}_V \cup \mathcal{C}_A \cup \mathcal{C}_E\}$:

- **Video cues** $\mathcal{C}_V$: visual behaviors such as facial expressions and body postures.

- **Audio cues** $\mathcal{C}_A$: paralinguistic descriptions such as intonation, rhythm, and volume.

- **Emotion cues** $\mathcal{C}_E$: inferential statements that explicitly connect the observed cues to the final emotion prediction.

For each ground-truth cue $c \in \mathcal{C}$, we obtain its embedding $\mathbf{e}(c)$ with Qwen3-Embedding (Zhang et al., 2025). Similarly, for each model generated output, we segment its intermediate thinking process into a sequence of clauses $\mathcal{U} = \{u_1, \ldots, u_K\}$ using punctuation-based splitting, and compute the embedding $\mathbf{e}(u_k)$ for each clause.

**Reward based on evidence coverage.** For each ground-truth cue $c$, we measure whether it is *covered by at least one clause* in the model's reasoning. Concretely, we first build an Evidence-Routing Matrix $\mathbf{M} \in \mathbb{R}^{|\mathcal{C}| \times K}$, where each entry

$$\mathbf{M}_{c,k} = \cos\big(\mathbf{e}(c), \mathbf{e}(u_k)\big) \tag{1}$$

is the cosine similarity between a ground-truth cue $c$ and a generated clause $u_k$. We then match each cue to its best supporting clause by taking the maximum similarity:

$$M(c) = \max_{k \in [K]} \mathbf{M}_{c,k}. \tag{2}$$

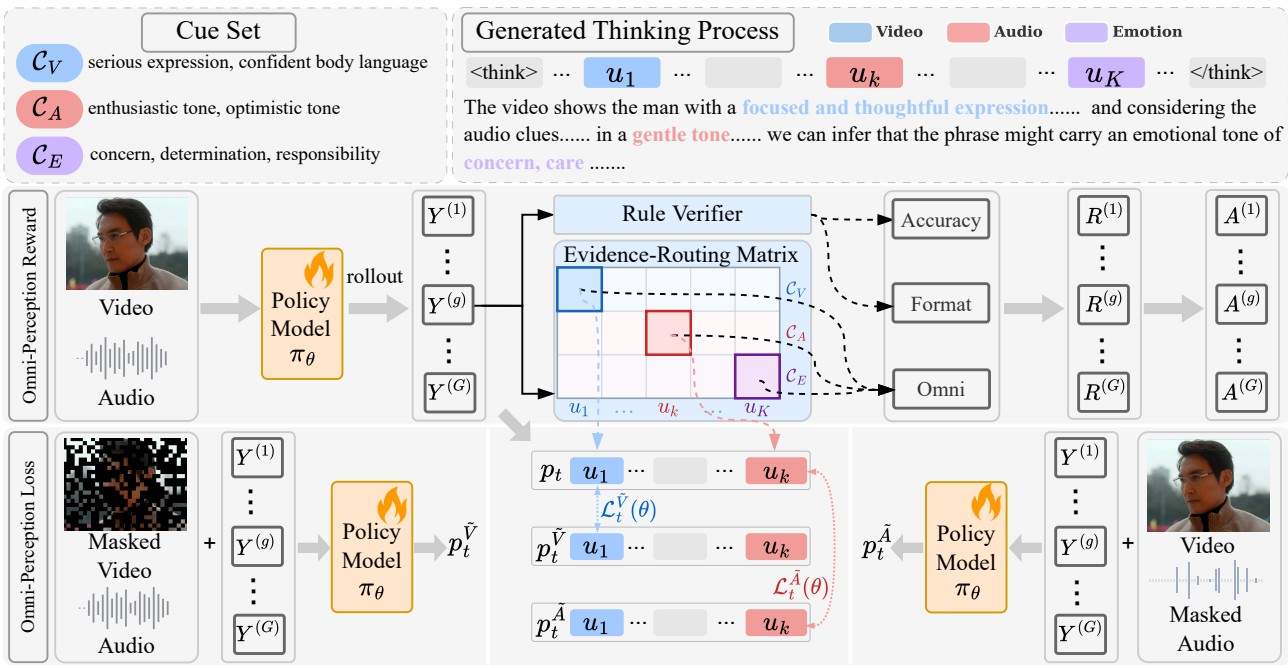

*Figure 2.* The framework of OPPO. Top: Fine-grained evidence is organized into video cues $\mathcal{C}_V$, audio cues $\mathcal{C}_A$, and emotion cues $\mathcal{C}_E$. CoT is segmented into clauses $\mathcal{U}$, which are identified as video, audio, or emotion types based on their semantic content. Middle: The Omni-Perception Reward (3.1). An Evidence-Routing Matrix computes the semantic similarities between generated clauses and ground-truth cues. This encourages the model to explicitly cover diverse multimodal evidence during reasoning. Bottom: The Omni-Perception Loss (3.2). We construct unimodal masked inputs and maximize the KL divergence between the full and masked distributions on modality-specific tokens $u_k$. This enforces faithfulness by penalizing the model for hallucinating information from missing modalities.

Finally, we convert this similarity into a normalized hit score

$$s(c) = \frac{1}{1-\delta} \max\big(M(c) - \delta, 0\big), \qquad (3)$$

where $\delta$ is a similarity threshold: if $M(c) \leq \delta$, the cue is treated as *unmatched* ($s(c) = 0$), and if $M(c) > \delta$, we linearly map the range $[\delta, 1]$ to $[0, 1]$ using the factor $\frac{1}{1-\delta}$. This way, only cues with sufficiently strong semantic matches contribute positively to the evidence coverage.

The omni-perception reward aggregates cue-level scores by averaging over the three cue sets:

$$R_{\text{omni}} = \sum_{\mathcal{C}' \in \{\mathcal{C}_V, \mathcal{C}_A, \mathcal{C}_E\}} \frac{1}{|\mathcal{C}'|} \sum_{c \in \mathcal{C}'} s(c), \qquad (4)$$

so that video, audio, and emotion cues are each rewarded according to the fraction of annotated cues that are covered. The overall reward used by GRPO augments the task and format rewards with this omni-perception term:

$$R_{\text{total}} = R_{\text{acc}} + R_{\text{fmt}} + \alpha \, R_{\text{omni}}, \qquad (5)$$

where $\alpha > 0$ controls the trade-off between end-task accuracy and cue-grounded reasoning.

### 3.2. Omni-Perception Loss

To enforce faithfulness, we introduce Omni-Perception Loss, which contrasts the policy's behavior under the full input

and a unimodal-masked counterfactual input on tokens corresponding to the masked modality. In other words, when a modality is removed, the model is encouraged to retract or weaken claims about that modality rather than hallucinating them from the remaining context.

**Unimodal masking and token-level discrepancy.** Given $X = \{V, A, T\}$, we construct counterfactual inputs by randomly masking a ratio $\rho$ of one modality to zero:

$$X^{\tilde{V}} = \{\tilde{V}, A, T\}, \quad X^{\tilde{A}} = \{V, \tilde{A}, T\}, \qquad (6)$$

where $\tilde{V}$ and $\tilde{A}$ denote the masked visual and acoustic streams, respectively. For each decoding step $t$, we write $p_t = \pi_\theta(\cdot \mid X, Y_{<t})$ for the token distribution under the full input, and denote

$$p_t^m = \pi_\theta(\cdot \mid X^m, Y_{<t}), \quad m \in \{\tilde{V}, \tilde{A}\}, \qquad (7)$$

the token distribution when modality $m$ is masked. We quantify the discrepancy between the full-input and masked-input distributions at step $t$ using the KL divergence:

$$\mathcal{L}_t^m(\theta) = \mathbb{D}_{\text{KL}}(p_t \,\|\, p_t^m), \quad m \in \{\tilde{V}, \tilde{A}\}. \qquad (8)$$

Intuitively, if $Y_t$ expresses modality-specific evidence of $m$, masking modality $m$ should substantially change the token distribution, yielding a large $\mathcal{L}_t^m(\theta)$. Conversely, if

*Table 1.* **Diagnostic Analysis on `MEP-Bench`.** We report **Utilization** (Evidence Recall) and **Faithfulness** (Perception Accuracy under masking). Note: HUMANOMNI-V2$^\ddagger$ serves as an *Oracle* (upper bound) recall reference.

| Model | Utilization | Faithfulness | |
|---|---|---|---|
| | Recall ($\uparrow$) | Acc@V ($\uparrow$) | Acc@A ($\uparrow$) |
| *SFT Baselines* | | | |
| AffectGPT | 49.14 | 35.00 | 26.00 |
| Qwen2.5-Omni | 53.00 | 57.00 | 33.80 |
| *RL Baselines & Oracle$^\ddagger$* | | | |
| AffectGPT-R1 | 54.85 | 54.20 | 30.00 |
| HumanOmni-V2$^\ddagger$ | 80.17$^\ddagger$ | 48.40 | 37.60 |
| *Ours* | | | |
| *Baseline* | 58.00 | 59.20 | 33.00 |
| **OPPO** | **70.44** | **76.40** | **50.60** |

the distribution remains similar after masking $m$, the model is likely relying on spurious cross-modal correlations to hallucinate content about the missing modality.

**Modality-specific token identification.** To identify which tokens should receive KL supervision, we match each clause against the omni-modal cue set $\mathcal{C}_O = \mathcal{C}_V \cup \mathcal{C}_A$. Reusing the Evidence-Routing Matrix $\mathbf{M}$ from Eq. (1), we match each clause $u_k$ to its best supporting cue in $\mathcal{C}_O$ by

$$c^*(u_k) = \arg\max_{c \in \mathcal{C}_O} \mathbf{M}_{c,k}, \qquad M(u_k) = \max_{c \in \mathcal{C}_O} \mathbf{M}_{c,k}. \quad (9)$$

We retain clause $u_k$ only when $M(u_k) > \delta$. For each retained clause, we assign it to a modality according to the type of its best-matching cue $c^*(u_k)$: clauses matched to $\mathcal{C}_V$ contribute their tokens to the visual set $\mathcal{T}_V$, whereas those matched to $\mathcal{C}_A$ contribute to the audio set $\mathcal{T}_A$.

**Overall objective.** Our Omni-Perception Loss encourages large KL divergence on the modality-specific token sets $\mathcal{T}_V$ and $\mathcal{T}_A$ identified above:

$$\mathcal{J}_{\text{omni}}(\theta) = \sum_{t \in \{\mathcal{T}_V, \mathcal{T}_A\}} \mathcal{L}_t^m(\theta) \quad (10)$$

Finally, we optimize GRPO with the overall reward $R_{\text{total}}$ in Eq. (5) and Omni-Perception Loss:

$$\mathcal{J}_{\text{total}}(\theta) = \mathcal{J}_{\text{GRPO}}(\theta) + \beta \mathcal{J}_{\text{omni}}(\theta), \quad (11)$$

where $\beta > 0$ controls the strength of regularization.

### 3.3. Comparison to Perception Enhancement Methods

We review prior methods that aim to improve perception in multimodal emotion reasoning and analyze their limitations. In Sec. 4, we further present empirical evidence consistently showing that these methods are less effective than our OPPO.

(1) *AffectGPT-R1 (Lian et al., 2025c) and HumanOmni-V2 (Yang et al., 2025):* These two models adopt an LLM-as-a-judge framework, where an LLM evaluates whether intermediate thoughts make better use of multimodal cues. Concretely, given a pair of reasoning chains, the LLM is asked to assign a higher reward to the one it judges as better. However, this design has two key limitations. First, the judgment is coarse-grained and heavily dependent on the quality of the judging LLM. In contrast, our Omni-Perception Reward explicitly extracts fine-grained evidence cues and encourages each cue to be grounded in the model's output; as reported in AffectGPT-R1, LLM-as-a-judge constraints can even disrupt reasoning, leading to lower cue recall and more grounding failures. Second, this reward cannot resolve the unfaithfulness issue, because it cannot distinguish whether a "better" reasoning chain is grounded in modality-specific cues or merely produced by hallucinating plausible-sounding explanations.

(2) *PAPO (Wang et al., 2026c):* PAPO aims to enhance perception by partially masking one modality and enforcing a KL divergence between the model's output distributions under the full and masked inputs. However, this design has two limitations. First, PAPO does not distinguish between different types of output tokens: the KL constraint is applied uniformly to every token, including those that are unrelated to the masked modality. As a result, many tokens are unnecessarily pushed away from their full-input distributions, making optimization unstable and highly sensitive to hyper-parameters. In contrast, our approach applies KL only to modality-specific evidence tokens identified via cue matching, resulting in more targeted supervision and substantially more stable optimization. Second, PAPO cannot reliably address cross-modal hallucination. Since the KL constraint is applied to *all* tokens, the model can satisfy it by globally perturbing its output distribution under masking, including tokens unrelated to the masked modality. Consequently, changes observed on modality-specific tokens are entangled with these global shifts and no longer indicate whether the statements truly depend on the masked modality.

## 4. Experiments

### 4.1. Setup

**Tasks and datasets.** Following prior work (Lian et al., 2025c), we adopt the comprehensive MER-UniBench (Lian et al., 2025a) for evaluation, which aggregates nine diverse multimodal emotion datasets covering three distinct tasks: (1) Fine-grained Emotion Recognition: OV-MERD+ (Lian et al., 2025a), (2) Basic Emotion Recognition: MER23 (Lian et al., 2023a), MER24 (Lian et al., 2024), MELD (Poria et al., 2019), IEMOCAP (Busso et al., 2008), and (3) Sentiment Analysis: MOSI (Zadeh et al., 2016), MOSEI (Zadeh et al., 2018), SIMS (Yu et al., 2020),

*Table 2.* **Main results on MER-UniBench.** This table presents a comprehensive performance comparison across sentiment analysis, basic emotion recognition, and fine-grained emotion recognition tasks. `OPPO` achieves state-of-the-art performance.

| Model | Sentiment Analysis | | | | Basic Emotion | | | | Fine | Mean |
|---|---|---|---|---|---|---|---|---|---|---|
| | MOSI | MOSEI | SIMS | SIMSv2 | MER23 | MER24 | MELD | IEMOCAP | OV-MERD+ | |
| Qwen-Audio (Chu et al., 2023) | 70.09 | 46.90 | 70.73 | 65.26 | 41.85 | 31.61 | 49.09 | 35.47 | 32.36 | 49.26 |
| SALMONN (Tang et al., 2024) | 81.00 | 67.03 | 68.69 | 65.93 | 55.53 | 45.38 | 45.62 | 46.84 | 45.00 | 57.89 |
| VideoChat2 (Li et al., 2024a) | 66.84 | 54.32 | 69.49 | 70.66 | 33.67 | 54.50 | 36.64 | 48.70 | 39.21 | 52.67 |
| LLaMA-VID (Li et al., 2024b) | 61.78 | 63.89 | 69.35 | 67.48 | 50.72 | 57.60 | 42.75 | 46.02 | 45.01 | 56.07 |
| Chat-UniVi (Jin et al., 2024) | 54.53 | 63.18 | 68.15 | 66.36 | 57.62 | 65.67 | 45.61 | 52.37 | 48.00 | 57.94 |
| mPLUG-Owl (Ye et al., 2023) | 72.40 | 72.91 | 72.13 | 75.00 | 56.86 | 59.89 | 49.11 | 55.54 | 48.18 | 62.45 |
| PandaGPT (Su et al., 2023) | 61.92 | 67.61 | 68.38 | 67.23 | 40.21 | 51.89 | 37.88 | 44.04 | 37.12 | 52.92 |
| R1-Omni (Zhao et al., 2025a) | 58.02 | 56.48 | 71.82 | 68.58 | 64.17 | 67.43 | 43.20 | 51.58 | 55.24 | 59.61 |
| Emotion-LLaMA (Cheng et al., 2024) | 66.13 | 67.66 | 78.32 | 77.23 | 59.38 | 73.62 | 46.76 | 55.47 | 52.97 | 64.17 |
| AffectGPT (Lian et al., 2025a) | 81.30 | 80.90 | **88.49** | 86.18 | 78.54 | 78.80 | 55.65 | 60.54 | 62.52 | 74.77 |
| AffectGPT-R1 (Lian et al., 2025c) | 79.65 | 80.18 | 87.26 | 85.75 | 84.51 | **93.13** | **66.71** | **74.26** | **68.39** | 79.98 |
| AffectGPT (Last Checkpoint) | 78.30 | 78.51 | 85.28 | 85.31 | 73.69 | 76.90 | 52.09 | 57.74 | 59.89 | 71.96 |
| AffectGPT-R1 (Reproduce) | 80.29 | 80.64 | 85.70 | 83.75 | 81.88 | 80.89 | 57.53 | 65.71 | 64.08 | 75.60 |
| **OPPO** | **86.50** | **84.63** | 86.22 | **88.26** | **87.73** | 90.34 | 64.06 | 73.60 | 67.16 | **81.05** |

*Table 3.* **Main results on MME-Emotion.** We report task-level CoT scores and overall recognition, reasoning, and CoT averages. `OPPO` achieves the best overall performance. We use `gemini-3.1-flash-lite-preview` as judge since GPT-4o is not available.

| Methods | ER-Lab | ER-Wild | FG-ER | FG-SA | ML-ER | Noise-ER | IR | SA | Mean | | |
|---|---|---|---|---|---|---|---|---|---|---|---|
| | | | | | | | | | Rec. | Rea. | CoT |
| PandaGPT (Su et al., 2023) | 28.5 | 23.2 | 25.2 | 41.4 | 28.9 | 26.3 | 29.2 | 37.7 | 21.6 | 38.4 | 30.0 |
| Emotion-LLaMA (Cheng et al., 2024) | 31.7 | 19.7 | 31.7 | 36.4 | 29.5 | 49.3 | 27.3 | 40.7 | 21.1 | 42.4 | 31.8 |
| AffectGPT (Lian et al., 2025a) | 35.0 | 32.8 | 32.6 | **44.2** | 31.3 | 50.7 | 30.0 | 47.9 | 19.7 | 57.4 | 38.6 |
| *Baseline* | 47.0 | 40.8 | 35.0 | 35.6 | 37.7 | 59.0 | 32.4 | 58.7 | 27.9 | 62.7 | 45.3 |
| **OPPO** | **54.6** | **45.5** | **42.0** | 40.7 | **39.6** | **66.2** | **34.8** | **60.6** | **31.0** | **68.1** | **49.5** |

SIMSv2 (Liu et al., 2022b). A detailed description of task definitions is provided in Appendix B. We further evaluate on MME-Emotion (Zhang et al., 2026a), which covers eight emotion-related tasks and reports recognition, reasoning, and CoT scores, providing a complementary evaluation of both answer correctness and reasoning quality.

**Baselines.** We adopt Qwen2.5-Omni (Xu et al., 2025a) as our backbone and reproduce AffectGPT-R1 (Lian et al., 2025c) following its official training recipe. Furthermore, we construct a stronger *Baseline* by adopting the optimized SFT/RL data split described in Section 3, trained via standard GRPO with only task and format rewards (see Appendix C.3 for more details). For fairness, all our reproduced and proposed models are evaluated using a single final checkpoint across all datasets. By contrast, prior work (Lian et al., 2025a;c) selects the best checkpoint per dataset, which tends to overestimate performance, see Appendix C.4 for more discussion.

**Implementation details.** In the cold-start SFT stage, we set the learning rate to 2e−5 and train for 2 epochs. During RL, we adopt a smaller learning rate of 2e−6 and train for 1 epoch with 3,262 overall training steps using GRPO-style sampling, where $G=4$ responses are generated per prompt. We fix the batch size to 1 with gradient accumulation steps set to 2 throughout training. The KL coefficient with respect to the reference model is set to $0.04$. For our `OPPO`, we use $\alpha = 0.5$ for the Omni-Perception Reward, $\beta = 8e-3$ for the Omni-Perception Loss, a masking ratio of $\rho = 0.7$, and a similarity threshold of $\delta = 0.5$, which together yield the optimal overall performance. A detailed hyperparameter analysis is provided in Section 4.4. All experiments were conducted on 16 NVIDIA H100 GPUs.

### 4.2. Multimodal Emotion Perception Benchmark

**Benchmark construction.** We introduce `MEP-Bench`, a diagnostic benchmark derived from OV-MERD (Lian et al., 2023b), to quantify *utilization* and *faithfulness*. We filter cue-rich instances and obtain 300 samples. (1) *Utilization* is measured by cue recall, where Qwen2.5-7B-Instruct (Yang et al., 2024) decomposes human annotations and the generated thinking process into atomic cues that are subsequently embedded by Qwen3-Embedding; a ground-truth cue is re-

*Table 4.* **Ablation Study on MER-UniBench and `MEP-Bench`.** Building upon the Omni-Perception Reward $R_{\text{omni}}$, incorporating generic perception losses (*e.g.*, PAPO) negatively impacts performance due to unguided optimization. In contrast, our Omni-Perception Loss $\mathcal{J}_{\text{omni}}(\theta)$ yields consistent improvements across all metrics.

| Model Variant | Sentiment Analysis | | | | Basic Emotion | | | | Fine | Mean | Util. | Faithfulness | |
| --- | --- | --- | --- | --- | --- | --- | --- | --- | --- | --- | --- | --- | --- |
| | MOSI | MOSEI | SIMS | SIMSv2 | MER23 | MER24 | MELD | IEMOCAP | OV-MERD+ | | Recall | Acc@V | Acc@A |
| *Baseline* | 83.48 | **86.31** | 87.78 | 86.84 | 78.16 | 82.70 | 61.45 | 67.28 | 66.86 | 77.87 | 58.00 | 59.20 | 33.00 |
| + $R_{\text{omni}}$ | 84.83 | 85.04 | 86.48 | 86.95 | 85.40 | 85.90 | 63.37 | 70.03 | 67.09 | 79.45 | 62.55 | 64.80 | 39.40 |
| + PAPO | 83.47 | 85.66 | 86.21 | 87.36 | 86.27 | 87.09 | 63.28 | 69.12 | 65.65 | 79.34 | 58.29 | 62.00 | 34.60 |
| + PAPO-dual | 84.98 | 85.06 | **88.47** | 86.72 | 84.96 | 86.11 | 63.07 | 67.51 | 66.20 | 79.23 | 60.58 | 65.20 | 34.80 |
| + $\mathcal{J}_{\text{omni}}(\theta)$ | **86.50** | 84.63 | 86.22 | **88.26** | **87.73** | **90.34** | **64.06** | **73.60** | **67.16** | **81.05** | **70.44** | **76.40** | **50.60** |

*Table 5.* **Impact of Omni-Perception Reward Weight $\alpha$.** As $\alpha$ increases, we observe a progressively better balance between task performance and perception metrics. Notably, setting $\alpha \geq 0.6^{\dagger}$ leads to model format collapse in the later training stage.

| Weight $\alpha$ | Task Performance | | | | Util. | Faithfulness | |
| --- | --- | --- | --- | --- | --- | --- | --- |
| | Senti. | Basic. | Fine. | Mean | Recall | Acc@V | Acc@A |
| 0.2 | 86.21 | 75.58 | 66.97 | 79.35 | 59.16 | 61.20 | 39.20 |
| 0.3 | 84.96 | 75.98 | **67.52** | 79.03 | 58.63 | 63.80 | **40.40** |
| 0.4 | **86.22** | 73.75 | 66.08 | 79.14 | 60.53 | 61.80 | 39.20 |
| **0.5** | 85.82 | 76.17 | 67.09 | 79.45 | 62.55 | **64.80** | 39.40 |
| $0.6^{\dagger}$ | 85.43 | **76.77** | 66.79 | **79.51** | 63.09 | 64.40 | 39.00 |

*Table 6.* **Impact of Omni-Perception Loss Weight $\beta$.** Increasing $\beta$ to a sufficient level ($\geq$ 8e-3) effectively strengthens Faithfulness, while simultaneously yielding significant improvements in Utilization and overall task performance.

| Weight $\beta$ | Task Performance | | | | Util. | Faithfulness | |
| --- | --- | --- | --- | --- | --- | --- | --- |
| | Senti. | Basic. | Fine. | Mean | Recall | Acc@V | Acc@A |
| 2e-3 | **86.62** | 76.23 | 65.79 | 79.69 | 60.56 | 63.60 | 39.80 |
| 4e-3 | 86.60 | 76.59 | 66.34 | 79.90 | 61.53 | 70.60 | 40.80 |
| 6e-3 | 86.25 | 77.30 | 66.08 | 80.03 | 62.98 | 73.00 | 45.20 |
| 8e-3 | 86.40 | 78.93 | **67.16** | **81.05** | **70.44** | **76.40** | **50.60** |
| 1e-2 | 84.81 | **80.36** | 66.61 | 80.81 | 69.97 | 73.80 | 46.80 |

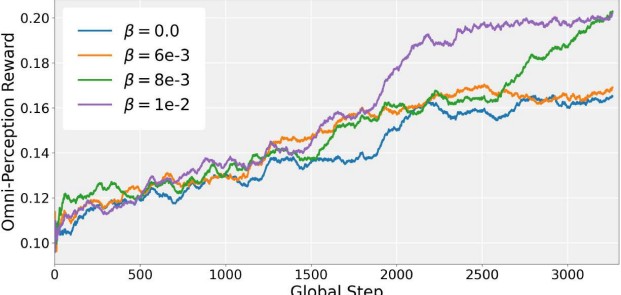

*Figure 3.* **Synergistic Effect of Omni-Perception Loss on Reward.** While minimal weights yield limited gains, increasing $\beta$ to a sufficient magnitude ($\geq$ 8e$-$3) significantly elevates the final Omni-Perception Reward.

called if its maximum cosine similarity to any generated cue exceeds 0.6. (2) *Faithfulness* is evaluated via a POPE-style unimodal masking probe (Li et al., 2023; Ye et al., 2026). On samples with strong audio–visual correlations, we construct 500 probe questions per modality targeting masked evidence (*e.g.*, mask video and ask "Does the video show a frown?"). Under masking, where the unmasked modality acts as a strong distractor, an affirmative answer signals hallucination, ensuring that correct refusals reflect genuine resistance to spurious priors.

**MEP-Bench results.** As shown in Table 1, existing models still struggle to achieve reliable perception under our two principles, *utilization* and *faithfulness*. For SFT baselines, performance remains limited: the fine-tuned Qwen2.5-

Omni (Xu et al., 2025a) recalls only 53.00% of human-annotated cues, and its low masking-based perception accuracy indicates frequent faithfulness violations. For RL baselines, AffectGPT-R1 (Lian et al., 2025c) brings only marginal recall gains and remains weak on faithfulness. HumanOmni-V2 (Yang et al., 2025) achieves oracle-level recall via OV-MERD cold-start training, yet still performs poorly on faithfulness probes, suggesting persistent cross-modal hallucination. In contrast, `OPPO` substantially improves both metrics: it boosts cue recall by 12.44% to 70.44%, narrowing the gap to the oracle reference, and increases perception accuracy by 17.20% on video and 17.60% on audio over the *Baseline*. These results demonstrate that `OPPO` effectively adheres to the principles of reliable perception, promoting both sufficient cue utilization and modality-grounded reasoning.

### 4.3. Main Results on MER-UniBench

**SoTA performance on MER-UniBench.** Table 2 presents the comprehensive evaluation results on MER-UniBench. `OPPO` achieves a new state-of-the-art mean score of 81.05%. Notably, it consistently outperforms the reproduced AffectGPT-R1 baseline across all datasets, with a substantial average improvement of 5.45%. Specifically, in Sentiment Analysis, `OPPO` attains superior accuracy on datasets like MOSI (86.50%) and SIMSv2 (88.26%), demonstrating its precision in discerning subtle variations in af-

*Table 7.* **Impact of Mask Ratio** $\rho$. We find that under relatively mild masking intensity, the model effectively enhances Faithfulness. Specifically, $\rho = 0.7$ achieves the optimal balance with Utilization and task performance.

| Ratio $\rho$ | Task Performance | | | | Util. | Faithfulness | |
|---|---|---|---|---|---|---|---|
| | Senti. | Basic. | Fine. | Mean | Recall | Acc@V | Acc@A |
| 0.6 | 85.46 | 79.61 | 66.63 | 80.77 | 68.94 | 75.20 | **55.20** |
| **0.7** | **86.40** | 78.93 | 67.16 | **81.05** | **70.44** | **76.40** | 50.60 |
| 0.8 | 85.38 | **79.67** | 66.58 | 80.75 | 68.92 | 73.60 | 50.20 |
| 0.9 | 86.13 | 78.66 | 65.76 | 80.55 | 69.32 | 64.20 | 40.80 |
| 1.0 | 85.27 | 78.81 | 66.75 | 80.34 | 67.53 | 69.60 | 48.20 |

*Table 8.* **Impact of Similarity Threshold** $\delta$. A relatively small $\delta$ approximates a modality-agnostic approach, causing a significant performance drop. For $\delta \geq 0.5$, increasing the threshold further improves Utilization, while $\delta = 0.5$ achieves the best trade-off.

| Thres. $\delta$ | Task Performance | | | | Util. | Faithfulness | |
|---|---|---|---|---|---|---|---|
| | Senti. | Basic. | Fine. | Mean | Recall | Acc@V | Acc@A |
| 0.40 | 86.24 | 76.40 | 66.85 | 79.71 | 60.73 | 68.40 | 46.40 |
| 0.45 | **86.69** | 76.24 | 66.59 | 79.81 | 60.62 | 68.20 | 44.60 |
| **0.50** | 86.40 | 78.93 | **67.16** | **81.05** | 70.44 | **76.40** | **50.60** |
| 0.55 | 80.18 | 79.76 | 67.16 | 80.76 | 71.77 | 72.00 | 48.00 |
| 0.60 | 85.58 | **80.03** | 66.93 | 81.04 | **72.54** | 68.00 | 46.60 |

fective polarity. In Basic Emotion Recognition, it delivers robust performance across diverse scenarios, particularly achieving 87.73% on MER23 and 90.34% on MER24, which highlights its strong categorical discrimination capabilities. Furthermore, in the challenging Fine-grained Emotion Recognition task, the model secures a high score of 67.16% on OV-MERD+, showcasing its effectiveness in open-vocabulary emotion prediction and accurately mapping complex signals onto the emotion wheels. It is worth noting that OPPO surpasses the reported results of Affect-GPT and AffectGPT-R1, even though they benefit from dataset-specific model selection. This highlights the superior robustness and effectiveness of our approach. We provide a detailed discussion regarding this evaluation setting in Appendix C.4.

**SoTA performance on MME-Emotion.** Table 3 reports the results on MME-Emotion. Since the original GPT-4o evaluator is no longer available, we uniformly re-evaluate all methods using `gemini-3.1-flash-lite-preview`. OPPO achieves the best overall performance, obtaining 31.0%, 68.1%, and 49.5% on the mean recognition, reasoning, and CoT metrics, respectively. Notably, OPPO improves the mean reasoning score from 62.7% to 68.1%, demonstrating a substantial enhancement in emotion reasoning capability. These results show that OPPO strengthens both affective reasoning and recognition, leading to superior CoT performance on MME-Emotion.

### 4.4. Ablation Studies and Analyses

**Overall ablation analyses.** Table 4 summarizes the contribution of each component. Adding the Omni-Perception Reward $R_{omni}$ to the *Baseline* consistently improves performance, increasing the mean score from 77.87% to 79.45% and cue recall from 58.00% to 62.55%. We further examine whether generic perception losses help by comparing two variants: (1) PAPO, which masks audio and video jointly and applies a sequence-level KL constraint, and (2) PAPO-dual, which separately masks audio and video to sum two KL terms, yet still applies constraints indiscriminately to all tokens. Both variants fail to improve task performance and

even degrade utilization and faithfulness, suggesting that unguided global constraints encourage the model to compensate via hallucinated evidence. In contrast, OPPO combines $R_{omni}$ with the proposed Omni-Perception Loss $\mathcal{J}_{omni}(\theta)$ and yields substantial gains across all metrics: it achieves the best mean score of 81.05%, boosts utilization to 70.44%, and improves faithfulness to 76.40% on video probes and 50.60% on audio probes.

**Impact of Omni-Perception Reward weight** $\alpha$. Table 5 studies the effect of the reward weight $\alpha$. As $\alpha$ increases from 0.2 to 0.5, cue recall steadily improves from 59.16% to 62.55%, indicating stronger multimodal evidence coverage. While $\alpha = 0.6$ brings only a marginal gain in mean score, it causes training instability and eventually triggers format collapse. We therefore set $\alpha = 0.5$ to balance stable optimization with sufficient cue utilization.

**Impact of Omni-Perception Loss weight** $\beta$. Table 6 analyzes the effect of the Omni-Perception Loss weight $\beta$. Increasing $\beta$ to 8e−3 enforces modality-grounded reasoning and leads to a sharp improvement in faithfulness, with probe accuracy reaching 76.40% on video and 50.60% on audio. This stronger grounding also synergizes with the reward signal, boosting cue recall to 70.44% and achieving the best overall performance. Further increasing $\beta$ to 1e−2 slightly degrades results, so we set $\beta = 8e−3$.

**Synergistic effect analysis.** To investigate the synergistic interaction between the Omni-Perception Loss and Reward, we visualize the reward training trajectories in Figure 3 by varying the weight $\beta$. We observe that with minimal weights ($\beta \leq 6e−3$), the reward yields limited gains. However, once $\beta$ reaches a sufficient magnitude ($\geq 8e−3$), the converged reward value increases substantially. This improvement arises because both components optimize the same underlying capability of multimodal perception, yet they are theoretically coupled: $\mathcal{J}_{omni}(\theta)$ expands the faithful information channel, which establishes the upper bound for $R_{omni}$. As derived in Appendix A, a sufficient $\beta$ is crucial to elevate this information ceiling, thereby unlocking the potential for maximizing the reward.

**Impact of mask ratio $\rho$.** Table 7 investigates sensitivity to the mask ratio $\rho$. The model benefits most from moderate masking, which ensures sufficient sensitivity to modality changes to enforce faithfulness. In contrast, increasing $\rho$ to 0.9 or 1.0 is overly aggressive and causes clear declines in both faithfulness and overall performance. We therefore set $\rho = 0.7$ as the final configuration, achieving the best balance between perception metrics and task performance.

**Impact of similarity threshold $\delta$.** Table 8 analyzes the effect of the similarity threshold $\delta$. Lower thresholds cause a notable performance drop, as the relaxed constraint becomes closer to a modality-agnostic setting like PAPO. Increasing $\delta$ tightens relevance, raising the standard of the reward and encouraging stronger cue retrieval, but it also reduces the tokens regulated by the Omni-Perception Loss and leads to sparser regularization. Therefore, we set $\delta = 0.5$, which provides the best balance across all metrics.

**Qualitative Analysis.** We provide comprehensive visualizations of reasoning trajectories in Appendix F. We observe that `OPPO` significantly enhances *utilization* by capturing richer multimodal cues compared to the baseline, directly leading to accurate predictions. Furthermore, regarding *faithfulness*, while the baseline driven by strong audio-video spurious correlations naturally generates false explanations, our model effectively mitigates such errors, providing strictly grounded reasoning. These improvements are consistent across fine-grained emotion recognition, basic emotion recognition, and sentiment analysis tasks.

## 5. Conclusion

In this work, we tackle unreliable omni-modal perception in Multimodal Emotion Reasoning. With our diagnostic benchmark `MEP-Bench`, we show that existing emotion-oriented Omni-MLLMs often underuse multimodal cues and produce unfaithful reasoning under cross-modal spurious correlations. To address this, we propose `OPPO`, an RL framework that combines Omni-Perception Reward to improve cue coverage with a targeted Omni-Perception Loss that penalizes hallucinations under unimodal masking. Experiments show that `OPPO` achieves state-of-the-art results on MER-UniBench while substantially improving both cue utilization and reasoning faithfulness. Overall, our findings highlight that reliable perception is essential for robust multimodal emotion reasoning.

## Acknowledgments

This work was supported by SenseTime Research, the National Natural Science Foundation of China (NSFC) under Grants U22A2094 and 62402471, and the Yangtze River Delta Science and Technology Innovation Community Joint Research (Basic Research) Project under Grant 2025CSJZN01600.

## Impact Statement

This paper presents work aimed at enhancing the robustness and reliability of Multimodal Emotion Reasoning by optimizing how models perceive and utilize diverse sensory signals. By ensuring that emotional interpretations are more consistently and accurately grounded in visual and acoustic evidence, our `OPPO` provides a foundation for more dependable affective computing systems. Such advancements are crucial for fostering trust and safety in human-AI collaboration, particularly in fields like assistive technologies, empathetic interface design, and supportive education. Ultimately, these improvements facilitate more natural and effective interactions, ensuring that artificial agents can better understand and respond to human emotional states in a stable and interpretable manner.

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

# A. Theoretical Analysis of Optimization Synergy

In this section, we provide an information-theoretic (Shannon, 1948) interpretation for the synergy between the Omni-Perception Loss $\mathcal{J}_{\mathrm{omni}}$ and the Omni-Perception Reward $R_{\mathrm{omni}}$. We follow the notations in the main paper: the multimodal input $X = \{V, A, T\}$, where $V$, $A$, and $T$ denote visual, acoustic, and textual streams, respectively. The policy $\pi_\theta$ generates a reasoning chain $Y$. We denote by $\mathcal{T}_m$ the set of evidence tokens for a target modality $m \in \{V, A\}$, and by $\mathcal{C}_m$ the corresponding ground-truth evidence cues (e.g., $\mathcal{C}_V$ or $\mathcal{C}_A$). For clarity, we denote the masked input where $m$ is removed as $X^{\tilde{m}}$ (e.g., if $m = V$, then $X^{\tilde{V}} = \{\tilde{V}, A, T\}$).

## A.1. Assumptions

**Assumption A.1** (Deterministic GT Evidence). The ground-truth evidence $\mathcal{C}_m$ is a deterministic function of the modality $m$, i.e., $\mathcal{C}_m = f(m)$. Moreover, $\mathcal{C}_m$ is conditionally independent of the remaining context $X^{\tilde{m}}$ given $m$.

This assumption captures the data generation process where human-annotated evidence cues are derived exclusively from the corresponding sensory modality, independent of other modalities when conditioned on $m$.

**Assumption A.2** (Variational Approximation). The masked policy $\pi_\theta(\cdot|X^{\tilde{m}})$ provides a variational approximation to the true marginal distribution $P_\theta(\mathcal{T}_m|X^{\tilde{m}}) = \mathbb{E}_m[\pi_\theta(\mathcal{T}_m|X)]$.

From an information-theoretic perspective, we posit that the masked policy serves as a tractable variational approximation to the intractable marginal distribution over evidence tokens. The Omni-Perception Loss implicitly encourages this approximation, and its empirical stability during training validates this assumption.

## A.2. Proposition: Omni-Perception Loss as a Conditional Mutual Information Proxy

We first recall the definition of conditional mutual information in our context. Given the random variables $\mathcal{T}_m$ (evidence tokens) and $m$ (target modality), conditioned on the masked context $X^{\tilde{m}}$, the conditional mutual information induced by the policy $\pi_\theta$ is defined as:

$$I_\theta(\mathcal{T}_m; m|X^{\tilde{m}}) = \mathbb{E}_X\left[\mathbb{D}_{\mathrm{KL}}\left(\pi_\theta(\cdot|X) \,\|\, P_\theta(\cdot|X^{\tilde{m}})\right)\right], \tag{12}$$

where $P_\theta(\cdot|X^{\tilde{m}})$ is the true marginal distribution of evidence tokens given the masked context. This quantity measures the information that $\mathcal{T}_m$ carries about $m$ when $X^{\tilde{m}}$ is observed.

**Proposition A.3** (Conditional Mutual Information Decomposition). *The Omni-Perception Loss $\mathcal{J}_{\mathrm{omni}}(\theta)$ can be decomposed into the conditional mutual information $I_\theta(\mathcal{T}_m; m|X^{\tilde{m}})$ and a variational approximation error:*

$$\mathcal{J}_{\mathrm{omni}}(\theta) = I_\theta(\mathcal{T}_m; m|X^{\tilde{m}}) + \Delta_{\mathrm{var}}(\theta), \tag{13}$$

*where* $\Delta_{\mathrm{var}}(\theta) = \mathbb{E}_{X^{\tilde{m}}}\left[\mathbb{D}_{\mathrm{KL}}\left(P_\theta(\mathcal{T}_m|X^{\tilde{m}}) \,\|\, \pi_\theta(\mathcal{T}_m|X^{\tilde{m}})\right)\right] \geq 0$.

*Proof.* By definition, the Omni-Perception Loss is the expected $\mathbb{D}_{\mathrm{KL}}$ divergence between the full policy and the masked policy on modality-specific tokens:

$$\mathcal{J}_{\mathrm{omni}}(\theta) = \mathbb{E}_X\left[\mathbb{D}_{\mathrm{KL}}\left(\pi_\theta(\mathcal{T}_m|X) \,\|\, \pi_\theta(\mathcal{T}_m|X^{\tilde{m}})\right)\right]. \tag{14}$$

By inserting $P_\theta$ and expanding the $\mathbb{D}_{\mathrm{KL}}$ as an expectation over $\mathcal{T}_m \sim \pi_\theta(\cdot|X)$, we obtain:

$$\begin{aligned}
\mathcal{J}_{\mathrm{omni}}(\theta) &= \mathbb{E}_X\left[\mathbb{E}_{y\sim\pi_\theta(\cdot|X)}\left[\log\frac{\pi_\theta(y|X)}{P_\theta(y|X^{\tilde{m}})} + \log\frac{P_\theta(y|X^{\tilde{m}})}{\pi_\theta(y|X^{\tilde{m}})}\right]\right] \\
&= \underbrace{\mathbb{E}_X\left[\mathbb{D}_{\mathrm{KL}}\left(\pi_\theta(\cdot|X) \,\|\, P_\theta(\cdot|X^{\tilde{m}})\right)\right]}_{I_\theta(\mathcal{T}_m;m|X^{\tilde{m}})} + \underbrace{\mathbb{E}_{X^{\tilde{m}}}\left[\mathbb{D}_{\mathrm{KL}}\left(P_\theta(\cdot|X^{\tilde{m}}) \,\|\, \pi_\theta(\cdot|X^{\tilde{m}})\right)\right]}_{\Delta_{\mathrm{var}}(\theta)}.
\end{aligned} \tag{15}$$

**Interpretation.** Proposition A.2 establishes $\mathcal{J}_{\mathrm{omni}}$ as a variational upper bound on the conditional mutual information. While maximizing this bound could theoretically increase the variational gap $\Delta_{\mathrm{var}}$ rather than the mutual information $I_\theta$, our framework prevents this trivial solution through parameter sharing. Since the variational approximation $\pi_\theta(\cdot|X^{\tilde{m}})$ and the full policy $\pi_\theta(\cdot|X)$ share the same backbone, they are structurally coupled. Arbitrarily inflating $\Delta_{\mathrm{var}}$ would require distorting the masked policy, which would inadvertently degrade the full policy. Consequently, maximizing $\mathcal{J}_{\mathrm{omni}}$ forces the model to genuinely increase the true mutual information, ensuring faithful modality-specific perception.

## A.3. Proposition: Information-Theoretic Upper Bound on Reward

To strictly connect the reward to our information-theoretic framework, we interpret the goal of $R_{\mathrm{omni}}$ as maximizing the information recovery of ground-truth cues.

**Information-theoretic interpretation of $R_{\mathrm{omni}}$.** We interpret the semantic alignment reward $R_{\mathrm{omni}}$ as an information-theoretic surrogate that encourages the recovery of ground-truth evidence cues from the generated modality-specific tokens. In particular, $R_{\mathrm{omni}}$ promotes semantic consistency between the generated evidence $\mathcal{T}_m$ and the ground-truth cues $\mathcal{C}_m$, which intuitively corresponds to reducing the conditional uncertainty of $\mathcal{C}_m$ given $\mathcal{T}_m$ and the masked context $X^{\tilde{m}}$, i.e., $H(\mathcal{C}_m \mid \mathcal{T}_m, X^{\tilde{m}})$. Since $I_\theta(\mathcal{T}_m; \mathcal{C}_m \mid X^{\tilde{m}}) = H(\mathcal{C}_m \mid X^{\tilde{m}}) - H(\mathcal{C}_m \mid \mathcal{T}_m, X^{\tilde{m}})$, and the conditional entropy $H(\mathcal{C}_m \mid X^{\tilde{m}})$ is independent of the model parameters $\theta$, encouraging the reduction of $H(\mathcal{C}_m \mid \mathcal{T}_m, X^{\tilde{m}})$ is consistent with increasing the conditional mutual information $I_\theta(\mathcal{T}_m; \mathcal{C}_m \mid X^{\tilde{m}})$. $R_{\mathrm{omni}}$ provides a practical and task-aligned proxy that steers the model toward evidence tokens that are semantically informative of the ground-truth cues.

**Proposition A.4** (Information Bound). *The mutual information between the generated evidence and the ground-truth cues is upper-bounded by the mutual information between the evidence and the modality:*

$$I_\theta(\mathcal{T}_m; \mathcal{C}_m | X^{\tilde{m}}) \leq I_\theta(\mathcal{T}_m; m | X^{\tilde{m}}). \tag{16}$$

*Proof.* We consider the chain rule of mutual information for the triple $(\mathcal{T}_m, m, \mathcal{C}_m)$ conditioned on $X^{\tilde{m}}$. We can decompose $I_\theta(\mathcal{T}_m; m, \mathcal{C}_m | X^{\tilde{m}})$ in two ways:

$$I_\theta(\mathcal{T}_m; m, \mathcal{C}_m | X^{\tilde{m}}) = I_\theta(\mathcal{T}_m; m | X^{\tilde{m}}) + I_\theta(\mathcal{T}_m; \mathcal{C}_m | m, X^{\tilde{m}}), \tag{17}$$

$$I_\theta(\mathcal{T}_m; m, \mathcal{C}_m | X^{\tilde{m}}) = I_\theta(\mathcal{T}_m; \mathcal{C}_m | X^{\tilde{m}}) + I_\theta(\mathcal{T}_m; m | \mathcal{C}_m, X^{\tilde{m}}). \tag{18}$$

Since $\mathcal{C}_m$ is a deterministic function of $m$ (Assumption A.1), given $m$, $\mathcal{C}_m$ provides no additional information about $\mathcal{T}_m$. Therefore, $I_\theta(\mathcal{T}_m; \mathcal{C}_m | m, X^{\tilde{m}}) = 0$. Equating the two decompositions, we have:

$$I_\theta(\mathcal{T}_m; m | X^{\tilde{m}}) = I_\theta(\mathcal{T}_m; \mathcal{C}_m | X^{\tilde{m}}) + \underbrace{I_\theta(\mathcal{T}_m; m | \mathcal{C}_m, X^{\tilde{m}})}_{\geq 0}. \tag{19}$$

Because mutual information is non-negative, it follows directly that:

$$I_\theta(\mathcal{T}_m; \mathcal{C}_m | X^{\tilde{m}}) \leq I_\theta(\mathcal{T}_m; m | X^{\tilde{m}}). \tag{20}$$

**Interpretation.** Proposition A.3 establishes a fundamental bottleneck: the amount of information that $\mathcal{T}_m$ can convey about $\mathcal{C}_m$ is strictly limited by the information that $\mathcal{T}_m$ extracts from the modality $m$ itself. Consequently, achieving a high reward is information-theoretically impossible without a sufficiently high mutual information $I_\theta(\mathcal{T}_m; m | X^{\tilde{m}})$. This justifies the role of $\mathcal{J}_{\mathrm{omni}}$ as a necessary regularizer that expands the upper bound (the channel capacity $I_\theta(\mathcal{T}_m; m | X^{\tilde{m}})$), thereby enabling the reward maximization.

## A.4. Discussion: Synergistic Mechanism and Empirical Validation

**Synergy mechanism.** Our theoretical analysis elucidates the synergistic collaboration between the Omni-Perception Loss $\mathcal{J}_{\mathrm{omni}}$ and the Omni-Perception Reward $R_{\mathrm{omni}}$. The optimization process relies on their distinct yet coupled functions: while Propositions A.2 and A.3 establish the information-theoretic bounds, we specifically analyze the optimization dynamics as follows. First, $\mathcal{J}_{\mathrm{omni}}$ serves to expand the faithful information channel by addressing the information theory derived in Proposition A.3. It provides a dense, token-level supervision signal that raises the upper bound of faithful information, ensuring the model possesses sufficient capacity to transmit modality features. Second, from a dynamics standpoint, this dense signal mitigates the optimization variance stemming from the sparse reward $R_{\mathrm{omni}}$. It steers the policy search space toward modality-sensitive regions, discouraging collapse into hallucinated local optima before the sparse reward signal becomes effective. Complementarily, $R_{\mathrm{omni}}$ encourages the utilization of semantically aligned cues. Acting as a steering mechanism for precision, it transforms this raw capacity into task-specific utility by guiding the model to filter noise and leverage evidence that is discriminative for emotion understanding.

**Empirical Validation.** This theoretical dependency is directly validated by the observations in Figure 3 and Table 6. The hyperparameter $\beta$ explicitly controls the weight of the Omni-Perception Loss, thereby determining the strength of the signal

used to expand the information capacity. Specifically, when $\beta$ is insufficient, the loss signal is too weak to counteract the high variance of the RL gradients or drive the extraction of modality features; consequently, the mutual information remains low, rendering the reward optimization ineffective due to the strict upper bound on performance. Conversely, with an appropriate $\beta$, the increased weight ensures that the dense supervision successfully reduces optimization variance and elevates this information-theoretic ceiling. This stabilized landscape paves the way for the reward to become effective, enabling the model to efficiently capture discriminative cues and achieve high performance.

## B. Tasks Details

We evaluate our method on MER-UniBench (Lian et al., 2025a), a unified benchmark that aggregates nine widely used multimodal emotion datasets spanning three representative tasks: *Fine-grained Emotion Recognition*, *Basic Emotion Recognition*, and *Sentiment Analysis*. This appendix provides a self-contained description of task definitions, datasets, and evaluation metrics.

### B.1. Emotion Wheel-based Evaluation Metric

Multimodal emotion reasoning outputs free-form emotion words with substantial lexical variability (*e.g.*, inflections and synonyms), making exact string matching unreliable. Following prior work (Lian et al., 2025a;c), we adopt an *Emotion Wheel (EW)-based set-level* metric and apply a three-level normalization to both predictions and ground-truth labels.

**Level 1.** Different morphological forms are mapped to their base form, *e.g.*, *happier* and *happiness* are mapped to *happy*. This mapping function is denoted as $F_{l_1}(\cdot)$.

**Level 2.** Semantically equivalent emotion words are mapped to a unified canonical form, *e.g.*, *joyful* and *happy*. This function is denoted as $F_{l_2}(\cdot)$.

**Level 3.** Emotion wheels provide a structured organization of emotions, where fine-grained emotions are arranged as outer labels and basic emotions as inner categories. We adopt $K = 5$ emotion wheels following prior work (Lian et al., 2025a), as illustrated in Figure 4. For each wheel $w_k$, all outer labels are mapped to their corresponding inner categories using $F_{l_3}^{w_k}(\cdot)$.

The complete grouping function is defined as:

$$G_{w_k}(\cdot) = F_{l_3}^{w_k}\big(F_{l_2}(F_{l_1}(\cdot))\big), \quad k \in [1, K]. \tag{21}$$

**Set-level metric.** Since both predicted and ground-truth emotion labels are variable-length sets, we employ set-level evaluation. Suppose the dataset contains $N$ samples. For sample $i$, let the ground-truth emotion set be $\mathcal{Y}_{(i)} = \{y_{(i)}^j\}_{j=1}^{n_i}$. Let $Y_{(i)}$ be the structured output generated by $\pi_\theta(\cdot \mid X_{(i)})$, and let $Y_{(i)}^a$ denote the content extracted from the <answer> field. We parse $Y_{(i)}^a$ into a predicted emotion set $\hat{\mathcal{Y}}_{(i)} = \{\hat{y}_{(i)}^j\}_{j=1}^{\hat{n}_i}$, with duplicate emotion words removed.

For each emotion wheel $w_k$, we define:

$$\text{Precision}_s^k = \frac{1}{N} \sum_{i=1}^{N} \frac{\left|G_{w_k}(\mathcal{Y}_{(i)}) \cap G_{w_k}(\hat{\mathcal{Y}}_{(i)})\right|}{\left|G_{w_k}(\hat{\mathcal{Y}}_{(i)})\right|}, \tag{22}$$

$$\text{Recall}_s^k = \frac{1}{N} \sum_{i=1}^{N} \frac{\left|G_{w_k}(\mathcal{Y}_{(i)}) \cap G_{w_k}(\hat{\mathcal{Y}}_{(i)})\right|}{\left|G_{w_k}(\mathcal{Y}_{(i)})\right|}, \tag{23}$$

$$\text{F}_s^k = 2 \times \frac{\text{Precision}_s^k \times \text{Recall}_s^k}{\text{Precision}_s^k + \text{Recall}_s^k}. \tag{24}$$

Finally, we compute the average F1-score across all emotion wheels as the final EW score:

$$\text{EW}(\mathcal{Y}_{(i)}, \hat{\mathcal{Y}}_{(i)}) = \frac{1}{K} \sum_{k=1}^{K} \text{F}_s^k. \tag{25}$$

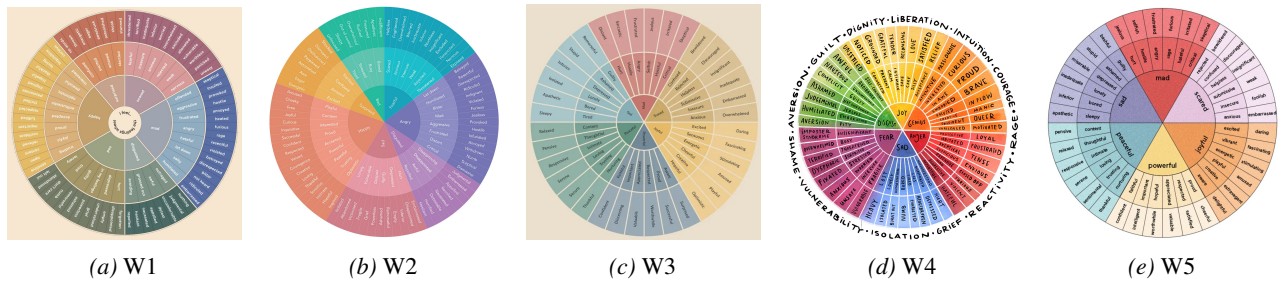

| *(a)* W1 | *(b)* W2 | *(c)* W3 | *(d)* W4 | *(e)* W5 |

*Figure 4.* **Emotion wheels.** We adopt five emotion wheels (W1–W5) to compute the EW-based metrics.

## B.2. Fine-grained Emotion Recognition

Fine-grained Emotion Recognition aims to predict a set of fine-grained emotion descriptors beyond basic categories. The output space is open-vocabulary, and each sample may correspond to multiple emotion labels.

We evaluate this task on OV-MERD+ (Lian et al., 2025a), which extends OV-MERD (Lian et al., 2023b). OV-MERD provides multimodal inputs with human-annotated fine-grained emotion descriptions, whereas the newly added samples in OV-MERD+ only include the required emotion labels.

**Evaluation metric.** Evaluation follows the MER-UniBench protocol (Lian et al., 2025a) and reports the EW-based set-level F1-score (defined above) as the primary metric.

## B.3. Basic Emotion Recognition

Basic Emotion Recognition focuses on predicting a single emotion label from a predefined set of basic emotions. We evaluate this task on MER2023 (Lian et al., 2023a), MER2024 (Lian et al., 2024), IEMOCAP (Busso et al., 2008), and MELD (Poria et al., 2019). Each sample is annotated with a single majority-voted basic emotion label.

**Evaluation metric.** Since the model produces free-form emotion words while each dataset provides one basic label, MER-UniBench evaluates this task with the *Hit Rate (HIT)*. For a sample with ground-truth label $y_{(i)}$ and predicted set $\hat{\mathcal{Y}}_{(i)}$, HIT is defined as:

$$\text{HIT} = \frac{1}{N} \sum_{i=1}^{N} \mathbb{I}\big[G_{w_k}(y_{(i)}) \in G_{w_k}(\hat{\mathcal{Y}}_{(i)})\big], \tag{26}$$

where $\mathbb{I}[\cdot]$ is the indicator function. A prediction is counted as correct if the ground-truth basic emotion is covered by the predicted set after EW-based normalization.

## B.4. Sentiment Analysis

Sentiment Analysis aims to predict sentiment polarity. We evaluate this task on CMU-MOSI (Zadeh et al., 2016), CMU-MOSEI (Zadeh et al., 2018), CH-SIMS (Yu et al., 2020), and CH-SIMS v2 (Liu et al., 2022b). The original annotations are continuous-valued sentiment scores. Following the MER-UniBench protocol, we binarize labels by mapping scores greater than zero to *positive* and scores less than zero to *negative*.

**Evaluation metric.** We report the Weighted Average F-score (WAF) as the primary metric due to label imbalance, consistent with MER-UniBench. Following the same multi-step evaluation protocol as AffectGPT (Lian et al., 2025a), we further convert the predicted emotion words into a sentiment polarity label using Qwen2.5-7B-Instruct. Specifically, given the extracted emotion words, the model is prompted to select the most likely sentiment category from a fixed candidate set (*e.g.*, {*positive*, *negative*, *neutral*}). The prompt used for this mapping is shown below.

---

**Sentiment Mapping Prompt**

*Please assume the role of an expert in emotions. We provide a set of emotion words describing a character. Please choose the most likely sentiment from the candidates: [positive, negative, neutral].*

---

# C. Implementation and Experiment Details

## C.1. Details of Task Reward

Multimodal Emotion Reasoning is formulated as an open-vocabulary generation problem. Since the label space is not fixed, the model may produce semantically equivalent emotion words with different surface forms, making exact string matching unreliable. Consistent with AffectGPT-R1 (Lian et al., 2025c), we define the task reward as an Emotion Wheel (EW)-based score computed between the predicted emotions and the ground-truth labels. Let $Y_{(i)}$ denote the generated structured sequence, and let $Y_{(i)}^a$ be the content extracted from the <answer> field. Given the ground-truth emotion labels $y_{(i)}$, the task reward is defined as:

$$R_{\mathsf{acc}} = \mathrm{EW}(Y_{(i)}^a, y_{(i)}) \tag{27}$$

where $\mathrm{EW}(\cdot)$ is the set-level F1 score averaged over multiple emotion wheels, as specified in Appendix B.1. This reward provides a continuous signal to optimize open-vocabulary emotion predictions under synonym and lexical variation.

## C.2. Details of Format Rewards

To ensure structurally valid outputs, we additionally use a binary format reward $R_{\mathsf{fmt}}$. Recall that the model is required to generate a reasoning chain enclosed in <think>...</think> followed by the final prediction enclosed in <answer>...</answer>. We set:

$$R_{\mathsf{fmt}} = \begin{cases} 1, & \text{if } Y_{(i)} \text{ contains both required fields with valid tags,} \\ 0, & \text{otherwise.} \end{cases} \tag{28}$$

This reward only checks structural compliance, and helps stabilize RL training by discouraging generations that omit either the reasoning or the final prediction.

## C.3. Details of Baselines

We consider two baseline settings in our experiments:

**AffectGPT-R1.** The first baseline follows AffectGPT-R1 (Lian et al., 2025c), which applies GRPO-style reinforcement learning on top of a supervised warm-up stage. We reproduce this baseline on the Qwen2.5-Omni (Xu et al., 2025a) backbone by strictly following its official data usage and training recipe: it first performs SFT on the full MER-Caption+ dataset (Lian et al., 2025a), and then conducts RL on an additional 1k samples from MER2025-OV (Lian et al., 2025b).

**Our *Baseline*.** The second baseline is constructed by ourselves and differs from AffectGPT-R1 mainly in the SFT/RL data allocation. Instead of performing SFT on the full MER-Caption+ corpus, we randomly sample 5k instances from MER-Caption+ for SFT, and use all remaining MER-Caption+ samples for the subsequent GRPO-style RL stage. All other components, including the backbone architecture, reward definitions, and optimization algorithm, are kept identical.

Empirically, we find that allocating more training budget to RL is more effective for MER. Under the same dataset, increasing the number of RL steps leads to more consistent improvements in end-task performance than further extending the SFT stage. This motivates an RL-heavy schedule for our baseline, on top of which we later incorporate OPPO.

## C.4. Details of Experiment

**Experiment setting.** For fairness, we report results using a single final checkpoint for each model across all datasets, rather than selecting dataset-specific checkpoints. However, we note that some prior work (*e.g.*, AffectGPT (Lian et al., 2025a) and AffectGPT-R1 (Lian et al., 2025c)) reports results under a dataset-specific checkpoint selection protocol, where the best-performing checkpoint is chosen separately for each dataset. This protocol can lead to overly optimistic estimates because it effectively performs repeated model selection on the test set, which is not aligned with standard machine learning practice that treats test splits as strictly held-out for final evaluation. In contrast, our unified protocol provides a more faithful measurement of generalization and avoids implicit test-set tuning.

**Experiment results.** To facilitate a fair and direct comparison, Table 2 includes both: (i) the numbers reported in the original AffectGPT/AffectGPT-R1 papers under their dataset-specific protocol, and (ii) their performance re-evaluated under our unified setting, so that improvements can be compared under the same evaluation conditions. The impact of this protocol is

evident from the gap between AffectGPT and AffectGPT (Last Checkpoint), where per-dataset selection yields consistently higher numbers. In contrast, we evaluate using a single final checkpoint across all datasets. Under this unified setting, `OPPO` outperforms other models on each dataset. Moreover, even when compared to the dataset-specific results reported for AffectGPT-R1, `OPPO` still achieves a higher mean score, improving from 79.98% to 81.05%. These results suggest that our gains are not attributable to checkpoint selection, but reflect consistently strong performance across diverse datasets.

## D. Additional Experimental Analysis

**Comparison with LLM-as-a-judge.** We compare our embedding-based reward with LLM-as-a-judge alternatives. Unlike LLM judges that provide only detached scalar scores, our reward is tightly coupled with Omni-Perception Loss and supports token-level evidence routing. This is critical for suppressing cross-modal hallucination, since a better-looking reasoning chain may still rely on hallucinated rather than modality-grounded cues. Our method instead computes a continuous reward while constructing the Evidence-Routing Matrix for the KL penalty, enabling faithful modality-specific optimization.

*Table 9.* **Comparison with LLM-as-a-judge rewards.** We report downstream performance and reward computation time per sample.

| Reward | Senti. | Basic | Fine | Mean | Time(s) |
|---|---|---|---|---|---|
| Coarse Judge | **86.17** | 71.95 | **67.58** | 77.75 | 0.15 |
| Fine Judge | 84.97 | 74.63 | 66.14 | 78.28 | 0.35 |
| Ours | 85.83 | **76.18** | 67.09 | **79.45** | **0.07** |

Second, embedding similarity provides a more stable optimization signal. Recent studies have shown that LLM evaluators may suffer from rating indeterminacy and high variance, which can introduce instability during reward-based training. Our reward is deterministic, continuous, and directly tied to semantic alignment, avoiding the sampling and calibration uncertainty of LLM judges.

Third, our reward is empirically more effective and computationally more efficient. We conduct ablations with two LLM-as-a-judge variants. Both use Qwen3-Instruct-14B as the evaluator, deployed on two H100 GPUs with vLLM. The coarse-grained judge scores the whole reasoning chain from 1 to 10, while the fine-grained judge verifies each extracted cue individually. We train model variants using only these judge rewards. As shown in Table 9, our Omni-Perception Reward achieves the best overall mean score while requiring the least reward computation time per sample. These results confirm that the proposed embedding-based reward is not only more aligned with our token-level faithfulness objective, but also more suitable for efficient reward optimization.

# E. Prompt for Multimodal Evidence Extraction

---

**Multimodal Evidence Extraction Prompt**

You are an expert linguistic annotator for a Multimodal Emotion Recognition dataset.

**Task:** Analyze the provided 'Reasoning Text' (the content inside <think> tags). Your goal is to extract the **multimodal evidence** the model used to reach the conclusion.

**Extraction Rules:**

1. **Visual Cues:** Extract specific phrases describing facial expressions, body language, or scene details mentioned in the text.

   - Examples: "eyes closed", "serious expression", "looking down", "furrowed brows", "clenched fists"
   - Keywords to look for: "expression", "face", "body", "posture", "gesture", "looking", "eyes", "video clues"

2. **Audio Cues:** Extract specific phrases describing voice quality, tone, speed, or volume.

   - Examples: "shaky voice", "aggressive tone", "fast speaking speed", "trembling voice", "loud volume"
   - Keywords to look for: "voice", "tone", "audio", "sound", "speaking", "speech"

**Output Format (JSON):**
{
    "visual_cues": ["phrase 1", "phrase 2"],
    "audio_cues": ["phrase 1", "phrase 2"],
}

**Important:**

- Extract short, semantically complete phrases (2–6 words), not full sentences.

- Do not hallucinate. Only extract what is explicitly written in the input text.

- If a category is missing in the text, leave the list empty.

---

# F. Qualitative Analysis

In this section, we provide extensive visualizations of reasoning trajectories to demonstrate the superiority of OPPO over the baseline model across utilization, faithfulness, and three emotion understanding tasks. Compared to the baselines, OPPO captures richer multimodal cues and provides strictly grounded reasoning, which directly translates into superior performance in complex emotion recognition scenarios.

**Utilization.** Table 10 depicts a woman in a joyful state; however, AffectGPT-R1 suffers from "thinking inertia" by focusing exclusively on audio cues and overlooking crucial happy facial expressions. By failing to capture any key multimodal cues, it yields an incorrect prediction of "anxious", whereas OPPO successfully identifies the woman's smile and relaxed posture, resulting in an accurate emotional assessment. Table 11 presents a man in a happy state, where AffectGPT-R1 again fails to anchor its reasoning in the correct evidence, providing only a generic "positive" prediction. In contrast, our model effectively retrieves specific cues such as "laughter" and "active gestures" to produce a precise "happy" prediction.

**Faithfulness on audio.** Table 12 depicts a woman in a state of sadness. AffectGPT-R1 provides an incorrect answer to the probe question; due to severe spurious correlation hallucinations during training, the model not only answers "Yes" but also offers a detailed yet false explanation. It is evident that the model hallucinates audio information based on visual cues even when the audio is masked. Furthermore, under this stress test, the model labels the woman as sad in the probe response while its original reasoning chain identifies her facial expression as neutral, demonstrating internal inconsistency. In contrast, OPPO provides a clear, concise, and correct answer. Table 13 features a video with multiple characters where the primary speaker is a young girl. AffectGPT-R1 misidentifies the subject, leading to incorrect reasoning, predictions, and faithfulness test results, while continuing to exhibit hallucinations of auditory details based on visual information. Conversely, our model identifies the correct subject, providing accurate emotion predictions and a hallucination-free response.

**Faithfulness on video.** Table 14 depicts a woman in a state of surprise and joy. While AffectGPT-R1 provides near-accurate reasoning in its initial trajectory, it produces a contradictory response to the probe question, hallucinating that the individual in the video is experiencing "anger, frustration, or anxiety" due to audio-driven hallucination. Our `OPPO` provides the correct answer without any such hallucination. Table 15 features an angry girl; however, AffectGPT-R1 fails to utilize actual visual cues during its reasoning. It only associates the aggressive audio with an angry facial expression during the probe question, despite the girl's expression being restrained and neutral in reality. In contrast, `OPPO` objectively reflects these visual details in its reasoning process and correctly denies the hallucinated behavior.

**Fine-grained emotion recognition.** Table 16 depicts a relaxed and happy individual instructing a subordinate. AffectGPT-R1 overlooks the person's facial expressions and erroneously predicts "anxious" based solely on audio cues. In contrast, `OPPO` successfully retrieves visual cues such as the "relaxed posture and smile" to correctly predict "relaxed" and "happy". Table 17 primarily features an angry individual. While AffectGPT-R1 only predicts emotions related to "frustration", our model accurately identifies the specific emotion of "anger". Although both anger and frustration are negative emotions, they are distinct; these results demonstrate that `OPPO` is more precise in fine-grained emotion understanding.

**Basic emotion recognition.** Table 18 depicts a woman teasing her friend in a happy state. However, AffectGPT-R1 relies solely on the verbal content of the dialogue, leading to an incorrect prediction of "anxious". Our model, `OPPO`, correctly analyzes the multimodal cues to successfully predict "happiness". Table 19 showcases a sad man. AffectGPT-R1 erroneously interprets the mockery from others as the subject's own smile while overlooking his actual sad facial expression and tone, resulting in a false prediction of "happy". In contrast, `OPPO` provides the correct emotional assessment of "frustration" by objectively identifying the man's dissatisfaction.

**Sentiment analysis.** Table 20 showcases a confident woman delivering a presentation with positive sentiment. AffectGPT-R1 erroneously interprets her urgent vocal delivery as a sign of "anxious" emotion. In contrast, `OPPO` successfully recognizes through her facial expressions and body language that she is feeling comfortable and relaxed in her environment, leading to a correct prediction of "positive" sentiment. Table 21 features a man exhibiting positive but subtle sentiment. AffectGPT-R1 fails to capture the fleeting micro-expressions and is misled by the subject's cautious vocal tone, resulting in a conservative "neutral" prediction. In contrast, `OPPO` demonstrates superior sensitivity by successfully detecting these minute visual details, correctly identifying the positive sentiment.

*Table 10.* First example of utilization comparing AffectGPT-R1 with `OPPO`.

| Example of Utilization | |
|---|---|
| | 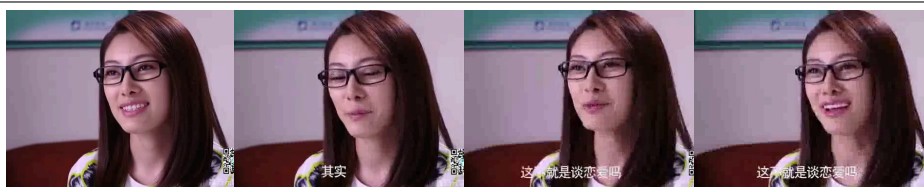 |
| Subtitle (En) | Actually, isn't this just dating? |
| Ground Truth | Labels: shy, relaxed, hesitant, joyful, friendly, happy
GT Evidence: slight smile on her face, mouth slightly upturned, eyes looking directly at the object, gaze moves downwards, pursing of the lips, gaze returns to the other person, smile becomes more apparent, soft and the tone is steady, longer pause between 'actually' and 'isn't this just dating?' |
| AffectGPT-R1 | `<think>`In the text, the caption reads: "Actually, isn't this dating?" This phrase might be a self-assessment or reaction from a woman to herself or someone nearby. Given the audio cues that include a personal voice filled with tension and anxiety, a high pitch, a rapid pace, and a slight tremble, we can infer that the tone of the statement might convey a sense of tension and excitement. Therefore, this sentence could be the woman expressing her excitement and nervousness about her own or someone else's romantic experience.`</think>`
`<answer>`excited, nervous, tense, anxious`</answer>`
Retrieved Evidence: *None* |
| OPPO | `<think>`In the text, the subtitle content says, "Actually, isn't this dating?" Based on the facial expression, body movements, tone, and subtitle content in the video, we can infer that the woman is experiencing a positive emotional state, possibly happiness or excitement. The woman's smile, relaxed posture, and the presence of the window in the background suggest that she is in a comfortable and pleasant environment, which further supports her positive emotions. The tone of the speaker in the audio is calm and steady, without any obvious signs of anxiety or nervousness. Based on the subtitle content, we can infer that the woman is talking about dating, which may be a topic she is excited about or looking forward to. Therefore, based on the facial expressions, body movements, tone, subtitle content, etc., in the video, we can infer that the woman is experiencing a positive emotional state, possibly happiness or excitement.`</think>`
`<answer>`happiness, excitement`</answer>`
Retrieved Evidence: woman's smile, relaxed posture, comfortable and pleasant environment, calm and steady tone |

*Table 11.* Second example of utilization comparing AffectGPT-R1 with `OPPO`.

| Example of Utilization | |
|---|---|
| | 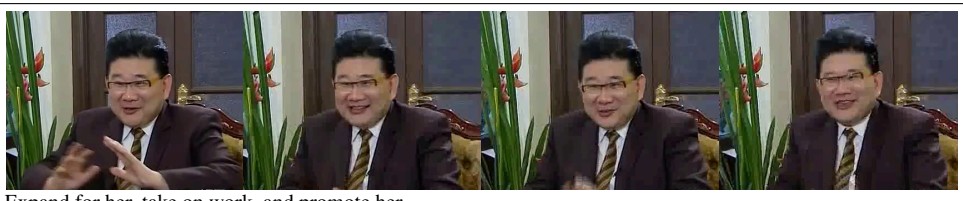 |
| Subtitle (En) | Expand for her, take on work, and promote her. |
| Ground Truth | Labels: happy, relaxed, hesitant, joyful, friendly, shy
GT Evidence: slight smile, eyes are slightly squinted, active gestures, obvious laughter |
| AffectGPT-R1 | `<think>`In the text, the caption reads: "This is about taking on jobs for her promotion." This sentence could be a man's discussion on a particular topic or making suggestions. Given the speaker's high-pitched, clear and concise tone, with a fast pace and energetic voice, we can infer that this sentence might be conveyed with an excited or passionate tone. Therefore, this sentence could be a man actively proposing ideas or suggestions, consistent with his overall positive mood.`</think>`
`<answer>`excited, passionate, positive`</answer>`
Retrieved Evidence: *None* |
| OPPO | `<think>`In the text, based on the facial expressions, body movements, and gestures of the man in the video, as well as the tone, pitch, and rhythm of the individual's voice in the audio clues, we can infer that this person is experiencing a positive emotional state, possibly feeling happy, excited, or passionate. The man's smile, laughter, and active gestures indicate that he is in a good mood and is enjoying himself. The audio clues suggest that the individual may be experiencing intense emotions, possibly due to excitement, joy, or passion. Therefore, this phrase "activate this promotion" could be the man's expression of enthusiasm and excitement about a certain activity or event.`</think>`
`<answer>`positive, happy, excited, passionate`</answer>`
Retrieved Evidence: man's smile, laughter, active gestures |

*Table 12.* First example of audio faithfulness comparing AffectGPT-R1 with `OPPO`.

| **Example of Audio Faithfulness** | |
| --- | --- |
| | 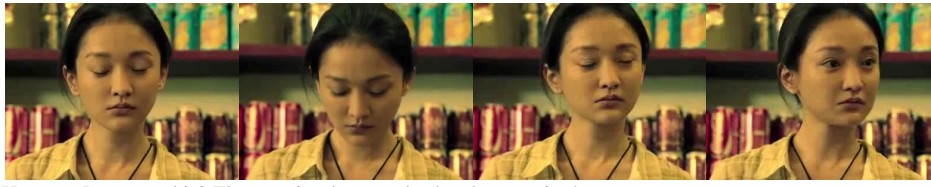 |
| Subtitle (En) | How can I manage this? That seat is where my husband was eating! |
| Ground Truth | Labels: questioning, angry, sad |
| AffectGPT-R1 | `<think>`In the text, caption content: "How do you calculate it, the person eating at that place is my husband." This sentence might be the woman expressing her confusion and surprise. Given the audio clues where describe the female voice as filled with frustration, annoyance, and impatience, and the video clues where the woman's facial expression is neutral, no signs of pain, and relaxed body language, we can infer that the woman may feel puzzled and astonished. She might not realize that the person eating at that place is her husband, which could lead to confusion and surprise. Therefore, this sentence expresses the woman's confusion and surprise.`</think>` `<answer>`confused, surprised, frustrated, annoyed, impatient`</answer>` Probe Question: Does the audio contain hint of crying? Model Answer: Yes, the audio contains hints of crying. The woman's expression and body language in the video suggest that she is experiencing a range of emotions, including sadness. |
| OPPO | `<think>`In the text, the subtitle reads: "How can I do this? The place where my husband used to sit and eat." Based on the facial expression, body language, and background of the woman in the video, as well as the trembling voice, fast-paced speech, and urgency in the audio, we can infer that this sentence may carry an emotion of sadness, frustration, or anxiety. The woman's facial expression and body language suggest that she may be experiencing a negative emotion, while the trembling voice, fast-paced speech, and urgency in the audio indicate that she may be feeling anxious or frustrated. Therefore, this sentence may be the woman's confusion, disappointment, or dissatisfaction regarding her husband's behavior or change.`</think>` `<answer>`sadness, frustration, anxiety`</answer>` Probe Question: Does the audio contain hint of crying? Model Answer: No. |

*Table 13.* Second example of audio faithfulness comparing AffectGPT-R1 with `OPPO`.

| **Example of Audio Faithfulness** | |
| --- | --- |
| | 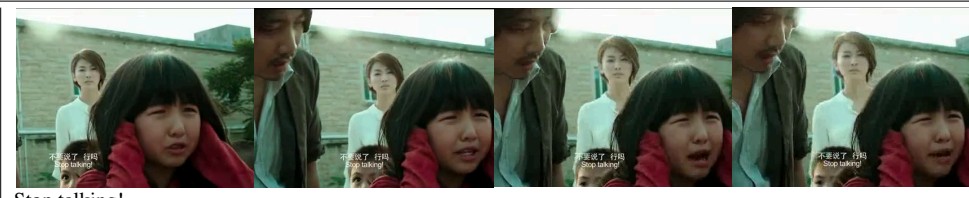 |
| Subtitle (En) | Stop talking! |
| Ground Truth | Labels: pressure, breakdown, anxiety, pain |
| AffectGPT-R1 | `<think>`In the text, the caption reads: "Don't say anything, Qiao Ma." This phrase could be an instruction or request to a woman by a man. Based on the audio clues described as a high-pitched scream and the video clues showing the woman crying and the man concerned, we can infer that this sentence might be the man's response to the woman's distress. The man may wish for the woman to stop expressing her emotions so as to alleviate her discomfort or distress. This sentence might be the man's response to the woman's distress, indicating his care and concern for her.`</think>` `<answer>`concerned, caring, anxious`</answer>` Probe Question: Does the audio contain high volume? Model Answer: Yes. The audio contains high volume. It seems to be an anxious, frustrated, or aggressive tone. This alignss with the video clues where the woman is trying to comfort the crying girl. |
| OPPO | `<think>`In the text, based on the facial expressions, body language, and surrounding environment of the girl in the video clues, it can be inferred that she is experiencing a negative emotion, possibly due to crying or distress. This emotional state alignss with the content of the subtitles where the individual is shouting loudly, possibly experiencing strong negative emotions such as anger, frustration, or sadness. Therefore, it can be inferred that the individual in the subtitles is expressing dissatisfaction, anger, or frustration towards the girl.`</think>` `<answer>`anger, frustration, sadness`</answer>` Probe Question: Does the audio contain high volume? Model Answer: No. |

*Table 14.* First example of video faithfulness comparing AffectGPT-R1 with `OPPO`.

| Example of Video Faithfulness | |
| --- | --- |
| | 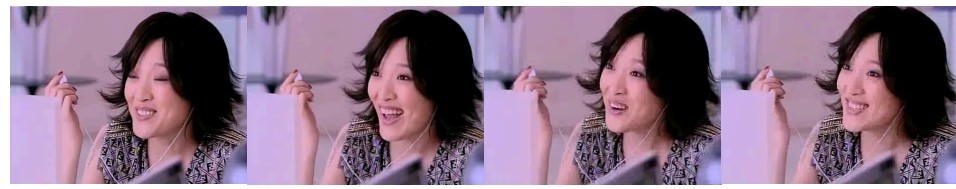 |
| Subtitle (En) | Director Bai, you're also paying attention to this matter! |
| Ground Truth | Labels: surprised, relaxed, positive, joyful, pleasant, happy |
| AffectGPT-R1 | `<think>`In the text, the caption reads: "Ms. White, you're also concerned about this." This sentence could be someone addressing Ms. White, either in inquiry or evaluation. Given the personal tone described as high and the fast pace, along with the woman's smile and positive mood depicted in the video clues, we can infer that this statement might carry an element of excitement or enthusiasm.`</think>` 
 `<answer>`excitement, enthusiasm, appreciation`</answer>` 
 Probe Question: Does the video show expression becoming more exaggerated? 
 Model Answer: Yes, the person's expression in the video is becoming more exaggerated, possibly due to heightened emotions such as anger, frustration, or anxiety. The tone of the voice in the audio is also becoming more intense, suggesting that the individual might be experiencing. |
| OPPO | `<think>`In the text, the caption reads: "Ms. Bai, you're also interested in this." Based on the facial expressions, body language, and smile of the woman in the video clues, as well as the high-pitched tone of the female voice in the audio clues, we can infer that this sentence might be a response or comment from the woman towards the Director. Given the overall positive mood and joyful demeanor of the woman, it's likely that the tone of this sentence could be one of excitement or anticipation. Therefore, this sentence might be the woman expressing her interest and excitement about a particular matter, which alignss with her overall positive emotional state.`</think>` 
 `<answer>`excited, anticipatory,joyful, positive`</answer>` 
 Probe Question: Does the video show expression becoming more exaggerated? 
 Model Answer: No. |

*Table 15.* Second example of video faithfulness comparing AffectGPT-R1 with `OPPO`.

| Example of Video Faithfulness | |
| --- | --- |
| | 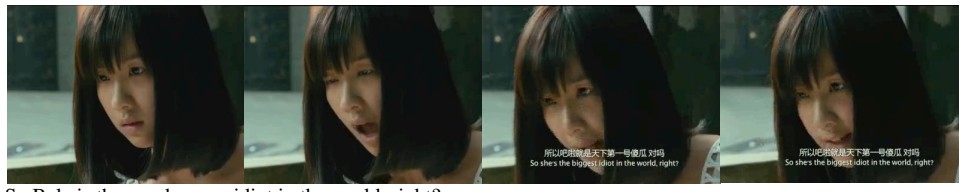 |
| Subtitle (En) | So Bala is the number one idiot in the world, right? |
| Ground Truth | Labels: excited, angry |
| AffectGPT-R1 | `<think>`In the text, the caption reads: "So, is Bala the number one fool in the world?" This sentence could be an evaluation or reaction by an individual towards someone or something. Given the audio cues that suggest the individual's voice is filled with anger and frustration, the tone is aggressive and confrontational, and the individual seems to be expressing a strong sense of power and dominance, we can infer that this sentence carries a negative judgment or an aggressive tone. The individual might be profoundly dissatisfied with someone or some action, leading to a strong reaction. Therefore, this sentence conveys the individual's anger and frustration.`</think>` 
 `<answer>`anger, frustration`</answer>` 
 Probe Question: Does the video show opens her mouth wide? 
 Model Answer: Yes, the woman's expression in the video is filled with anger and frustration, and her wide open mouth may be a strong reaction to a certain situation or event. At the same time, the audio clues describe the individual's voice as filled with anger. |
| OPPO | `<think>`In the text, the subtitle reads: "So Bala is the number one fool in the world, right?" Based on the description of the individual's high-pitched and agitated voice in the audio clues, as well as the woman's neutral expression and lack of obvious emotional cues in the video clues, we can infer that this sentence may carry a tone of anger or frustration. The individual may be expressing dissatisfaction or disagreement with someone or something. Therefore, this sentence may be a negative evaluation or accusation, aligning with the overall negative emotion displayed by the individual.`</think>` 
 `<answer>`anger, frustration`</answer>` 
 Probe Question: Does the video show opens her mouth wide? 
 Model Answer: No. |

*Table 16.* First example of fine-grained emotion reasoning comparing AffectGPT-R1 with `OPPO`.

| Example of Fine-grained Emotion Reasoning | |
|---|---|
| | 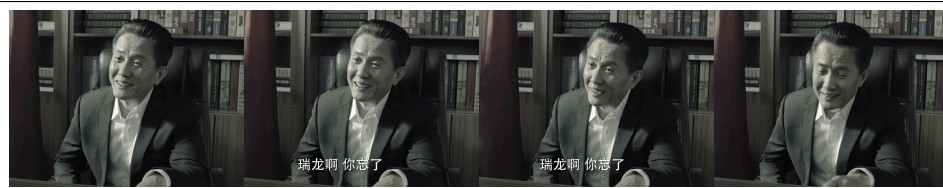 |
| Subtitle (En) | Li Lang, you forgot. |
| Ground Truth | emphasize, easy, joyful, friendly, happy, relax |
| AffectGPT-R1 | `<think>`In the text, the caption reads: "Lilang, you forgot." This sentence could be a reminder or inquiry from a man to someone or about something. Given the personal voice described in the audio cues as shaking, with a slow and deliberate pace, and considering the tone of the sentence, we can infer that this sentence might carry a sense of tension or anxiety. Therefore, it might be a reminder or inquiry from a man in a formal setting, consistent with his overall demeanor of confidence and self-assurance.`</think>` 
 `<answer>`confident, self-assured, tense, anxious`</answer>` |
| `OPPO` | `<think>`In the text, the subtitle content "Li Lang, you forgot" based on the facial expressions, body movements, tone, subtitle content, etc., in the video and audio clues, we can infer that this sentence may be a man's reminder or reminder to someone. Based on the man's relaxed posture and smile, as well as the comfortable and professional environment in the video, we can infer that this sentence may have a light-hearted or humorous tone. At the same time, based on the slight tremor in the individual's voice, slow and purposeful speech in the audio, we can infer that this sentence may carry a sense of urgency or importance. Therefore, this sentence may be the man reminding someone not to forget something important or reminding them to pay attention to something.`</think>` 
 `<answer>`relaxed, happy, humorous, urgent, important`</answer>` |

*Table 17.* Second example of fine-grained emotion reasoning comparing AffectGPT-R1 with `OPPO`.

| Example of Fine-grained Emotion Reasoning | |
|---|---|
| | 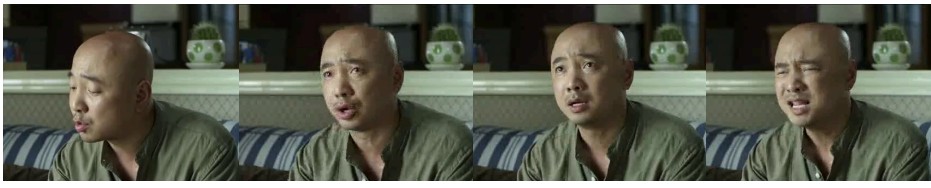 |
| Subtitle (En) | Gao Jianling turned back to look for her, but she refused. So what else can she do? |
| Ground Truth | angry, accuse |
| AffectGPT-R1 | `<think>`In the text, the subtitle content "Apologies, turn around to find her, she refused." Based on the video clues of the man's serious expression and body language, as well as the audio clues of the individual's voice trembling slightly and tone of uncertainty or stress, it can be inferred that this sentence may be the man expressing a sense of frustration, helplessness, or refusal. He may be dealing with a situation that makes him feel uncomfortable or unwilling, leading to a state of anxiety and pressure. This sentence may be him expressing his frustration or helplessness in dealing with this situation.`</think>` 
 `<answer>`frustration, helplessness, refusal, anxiety, pressure`</answer>` |
| `OPPO` | `<think>`In the text, caption content: "Gao Jianling turned back to find her, but she refused." Based on the facial expressions, body language, and posture of the man in the video clues, as well as the tone, pitch, and rhythm of the individual's voice in the audio clues, we can infer that this sentence may carry an emotion of disappointment, anger, or frustration. The man's facial expressions, body language, and posture indicate that he may be experiencing some negative emotions, while the individual's voice in the audio clues has a high pitch and fast pace, suggesting that they may be experiencing anxiety or stress. Therefore, this sentence may be a response to a refusal or rejection, expressing the individual's disappointment, anger, or frustration towards Gao Jianling.`</think>` 
 `<answer>`disappointment, anger, frustration`</answer>` |

*Table 18.* First example of basic emotion reasoning comparing AffectGPT-R1 with `OPPO`.

| Example of Basic Emotion Reasoning | |
|---|---|
| | 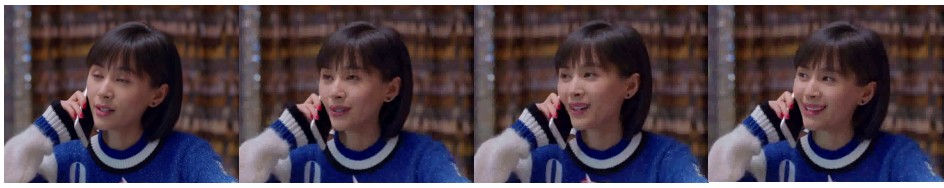 |
| Subtitle (En) | If you don't come up with a flawless lie, I won't help you with this. |
| Ground Truth | happy |
| AffectGPT-R1 | `<think>`In the text, the caption reads: "I won't help you unless you come up with a seamless lie." This sentence could be a demand or response from a woman over the phone. Given the audio clues that describe an individual's voice with high pitch and a rapid pace, along with a sense of urgency and frustration, we can infer that the tone might be tense or anxious. Therefore, this sentence might be the woman expressing her concerns or demands over the phone, which aligns with her overall demeanor of seriousness and concern.`</think>`
`<answer>`tense, anxious, serious, concerned`</answer>` |
| OPPO | `<think>`Based on the facial expressions, body movements, and tone of the individual described in the video clues, as well as the description of the individual's voice in the audio clues, we can infer that the individual is experiencing a positive emotional state, possibly happiness or excitement. The facial expressions and body movements in the video clues indicate that the individual is engaged in a pleasant conversation and is comfortable in their environment, while the audio clues describe the individual's voice as clear and bright, indicating that she is experiencing positive emotions such as happiness or excitement. Therefore, the phrase "If you don't come up with a flawless lie, I won't help you with this" in the subtitles may be a form of teasing or joking expression, which aligns with the overall positive emotions displayed by the individual.`</think>`
`<answer>`positive, happiness, excitement`</answer>` |

*Table 19.* Second example of basic emotion reasoning comparing AffectGPT-R1 with `OPPO`.

| Example of Basic Emotion Reasoning | |
|---|---|
| | 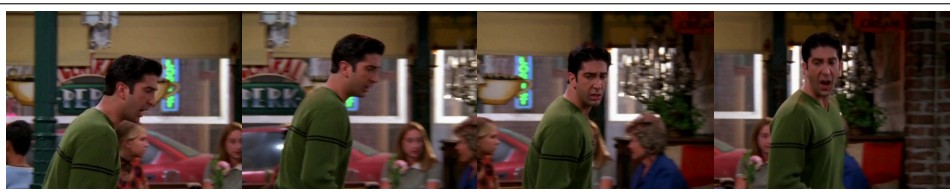 |
| Subtitle (En) | Fine! |
| Ground Truth | sadness |
| AffectGPT-R1 | `<think>`In the text, the caption content: "Gee!" This phrase could be a man's evaluation or reaction to a woman or a certain situation. Given the video hints of the man smiling and laughing, as well as the woman laughing along with him, it can be inferred that this phrase might be the man's evaluation of a woman or a certain situation, conveying his happiness and satisfaction. At the same time, based on the audio clues describing the man's voice filled with tension and excitement, it can be speculated that this phrase might be the man's evaluation of a situation that excites him or that he is looking forward to.`</think>`
`<answer>`happy, satisfied, excited`</answer>` |
| OPPO | `<think>`Based on the facial expressions, body language, and surrounding environment of the man in the video clues, as well as the audio clues indicating of laughter and possibly negative emotions, we can infer that this individual might be experiencing a negative emotional state, such as anger, frustration, or annoyance. The man's actions, facial expressions, and body language suggest that he is experiencing a negative emotional state, possibly due to dissatisfaction or annoyance with the current situation or event. The laughter in the audio clues may be a response from others to his emotional state, possibly indicating mockery or teasing. Therefore, this sentence "Fine!" might be the man expressing his dissatisfaction or annoyance, possibly with a tone of anger or frustration.`</think>`
`<answer>`anger, frustration, annoyance`</answer>` |

*Table 20.* First example of sentiment analysis comparing AffectGPT-R1 with `OPPO`.

| **Example of Sentiment Analysis** |
|---|

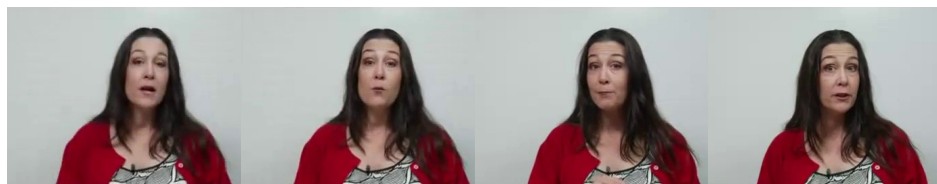

| Subtitle (En) | Okay, what happens at this point after we've taken this brief walk down memory lane, is the presentation of the gift. |
|---|---|
| Ground Truth | positive |
| AffectGPT-R1 | `<think>`In the text, the subtitle reads: "After this brief stroll down memory lane, it's time to present the gift." This sentence might be spoken by a woman in a video. Based on the audio cues described, which include a personal voice trembling, a quick pace, and a sense of urgency, along with the woman's expression of surprise or a strong emotional response in the video clues, we can infer that the tone of this sentence might be one of tension, excitement, or anxiety. The woman may be introducing an important or unexpected event, or about to present a gift. Therefore, this sentence could be the woman's expression of excitement and tension, which alignss with her overall emotional state.`</think>`
`<answer>`excited, tense, anxious`</answer>`
`<sentiment>`negative`</sentiment>` |
| OPPO | `<think>`Based on the facial expressions, body language, and overall demeanor of the woman in the video clues, as well as the description of the speaker's tone, intonation, and pace in the audio clues, we can infer that the woman may be experiencing a positive emotional state, such as happiness or excitement. The woman's facial expressions and body language suggest that she is feeling comfortable and relaxed in this environment, while the speaker's tone and intonation indicate that they are excited about the presentation of the gift and eager to share it with the audience. Therefore, based on the video and audio clues, we can infer that the woman is excited and happy about the presentation of the gift.`</think>`
`<answer>`excited, happy`</answer>`
`<sentiment>`positive`</sentiment>` |

*Table 21.* Second example of sentiment analysis comparing AffectGPT-R1 with `OPPO`.

| **Example of Sentiment Analysis** |
|---|

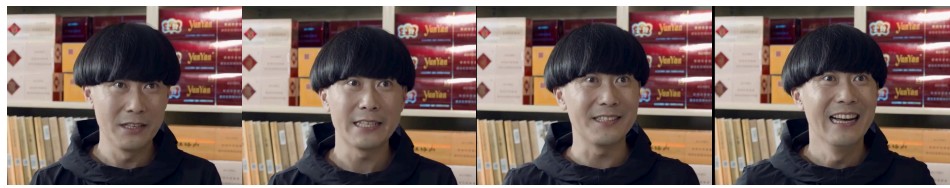

| Subtitle (En) | Don't worry, Aunt Wang, I'm very patient. |
|---|---|
| Ground Truth | positive |
| AffectGPT-R1 | `<think>`In the text, the caption reads: "Don't worry, Auntie Wang, I'm very patient." This sentence might be a man's comfort or assurance to Auntie Wang. Given the audio clues describing an individual with a clear, confident voice that also exhibits signs of anxiety and nervousness, along with the video clues showing the man in a relaxed posture with a neutral facial expression, we can infer that this statement might be the man's response to Auntie Wang's concerns or worry. He is trying to reassure her by expressing his patience and calmness, which alignss with his overall demeanor of relaxation and contentment.`</think>`
`<answer>`patient, calm, reassured, content, anxious, nervous`</answer>`
`<sentiment>`neutral`</sentiment>` |
| OPPO | `<think>`In the text, the subtitle says, "Don't worry, Aunt Wang, I am very patient." Based on the description of the individual's high pitch, excited tone, and the man's smile and laughter in the video clues, we can infer that this sentence is likely spoken by the individual in a positive emotional state, such as happiness or excitement. The individual may be reassuuring Aunt Wang that they are very patient, expressing their own positive emotions. This sentence may be the individual responding to Aunt Wang's concerns or doubts, indicating that they will handle the situation with care and understanding. Overall, this sentence aligns with the positive emotions displayed by the individual and the joyful atmosphere of the scene.`</think>`
`<answer>`positive, happiness, excitement, care, understanding`</answer>`
`<sentiment>`positive`</sentiment>` |

