# OpenReview forum: "Omni-Perception Policy Optimization for Multimodal Emotion Reasoning"
_ICML.cc/2026/Conference — ICML 2026 regular_

### Official Review · Reviewer_iiWR · 2026-02-27

**Soundness:** 3
**Presentation:** 3
**Significance:** 3
**Originality:** 3
**Overall Recommendation:** 4
**Confidence:** 4

**Summary:**

This paper proposes a novel reinforcement learning (RL) framework called Omni-Perception Policy Optimization (OPPO) to address a critical challenge in Multimodal Emotion Reasoning (MER): the lack of reliable and faithful multimodal perception.

**Compliance With Llm Reviewing Policy:**

Affirmed.

**Final Justification:**

I maintain my original score: weak accept.

**Key Questions For Authors:**

See weakness

**Limitations:**

yes

**Strengths And Weaknesses:**

Strength:

1. The proposed OPPO framework is elegant and directly addresses the diagnosed problems.
2. By focusing on improving the grounding and faithfulness of reasoning, this work tackles a core challenge for trustworthy and interpretable multimodal AI. The improvements are substantial and the method is likely to influence future work in multimodal reasoning beyond just emotion recognition.

Weakness:

1. The effectiveness of the Omni-Perception Reward hinges on the availability of high-quality, fine-grained "ground-truth" evidence cues. The paper uses GPT-5 for this extraction, which raises questions about the scalability, cost, and potential noise/bias introduced by this external model. An analysis of the sensitivity of OPPO to the quality of these extracted cues (e.g., using different LLMs or human annotations) would strengthen the robustness claim.
2. Since the paper aims to tackle the problem of emotion reasoning, rather than emotion recognition, I think the proposed method should be evaluated on emotion reasoning benchmarks, such as MME-Emotion.

---

> ### Author Rebuttal · Authors · 2026-03-31
>
> ## To Reviewer iiWR
> We sincerely thank the reviewer for the constructive feedback and recognition of our work. We address your insightful concerns regarding the cue extraction robustness and downstream reasoning evaluation below.
>
> **R1: Scalability and Robustness of Cue Extraction.** Extracting evidence (visual, audio, and emotion cues) from reasoning texts is a simple task without complex reasoning. Most modern large language models can perform this efficiently at scale with low cost. To rigorously evaluate OPPO's sensitivity to extraction quality, we compared GPT-5, Gemini-3.1-Flash-Lite-Preview, and DeepSeek-V3.1 from three perspectives:
> * **Human Validation:** We randomly sampled 200 instances and manually scored the extraction accuracy for visual, audio, and emotion cues, penalizing for missing or hallucinated information. As shown in Table 1, all models achieve over 94\% average accuracy with minimal variance, proving that high-quality extraction is not strictly bound to expensive proprietary models.
> * **Semantic Similarity:** We computed the pairwise cosine similarities of the extracted cues across these three models using Qwen-Embedding. As detailed in Table 2, the mean similarities across all modalities consistently exceed 0.90, demonstrating strong semantic alignment and minimal model-specific bias.
> * **Training Robustness:** To isolate the impact of different extraction models, we trained a model variant using only the Omni-Perception Reward guided by Gemini-extracted cues. The final performance shown in Table 3 is nearly identical to the GPT-5 counterpart. This demonstrates that our reward mechanism is highly resilient to minor extraction noise and is highly scalable using cost-effective alternative models.
>
> **Table 1: Human Evaluation of Cue Extraction Accuracy**
> |Model|Visual %|Audio %|Emotion %|Avg %|
> |-|-|-|-|-|
> |GPT-5|93.6|95.5|96.9|95.3|
> |Gemini-3.1|92.3|94.4|96.6|94.4|
> |DeepSeek-V3.1|92.5|94.0|96.3|94.2|
>
> **Table 2: Pairwise Semantic Similarity of Extracted Cues**
> |Model Pair|Visual|Audio|Emotion|Mean|
> |-|-|-|-|-|
> |Gemini vs. GPT-5|0.9137|0.9018|0.9547|0.9234|
> |Gemini vs. DeepSeek|0.9169|0.9004|0.9518|0.9230|
> |GPT-5 vs. DeepSeek|0.9170|0.9145|0.9526|0.9280|
>
> **Table 3: Task Performance with Different Extraction Models**
> |Extraction Model|Senti.|Basic.|Fine.|Mean|
> |-|-|-|-|-|
> |GPT-5 Original|85.83|76.18|67.09|79.45|
> |Gemini-3.1|86.37|75.75|66.42|79.43|
>
> **R2: Evaluation on MME-Emotion Benchmark.** We appreciate the constructive suggestion. Our paper demonstrates that enhancing the reasoning process inherently leads to better recognition performance. To further validate this on dedicated reasoning benchmarks, we conducted additional experiments on MME-Emotion. Due to recent OpenAI policy changes restricting access to the specific GPT-4o API version used in the original MME-Emotion paper, we utilized Gemini-3.1-Flash-Lite-Preview as the evaluator to ensure a fair and reproducible assessment. As shown in Table 4, OPPO substantially outperforms the baseline across all recognition and reasoning metrics. This confirms the effectiveness of our proposed method in advancing multimodal emotion reasoning capabilities. The experimental results will be added to the revised appendix. We are more than happy to discuss any further details during the interactive discussion phase.
>
> **Table 4: Performance on MME-Emotion Benchmark**
> |Method|Recognition|Reasoning|CoT|
> |-|-|-|-|
> |Baseline|27.9|62.7|45.3|
> |OPPO|31.0|68.1|49.5|

---

> > ### Author Rebuttal · Reviewer_iiWR · 2026-04-01
> >
> > I like to maintain my original score since it's already positive.

---

> > > ### Author Response · Authors · 2026-04-02
> > >
> > > Dear Reviewer,
> > >
> > > We sincerely thank you for your quick reply and your continued recognition of our work.
> > >
> > > Best regards,
> > >
> > > The Authors

---

### Official Review · Reviewer_wuw6 · 2026-03-08

**Soundness:** 2
**Presentation:** 3
**Significance:** 3
**Originality:** 2
**Overall Recommendation:** 4
**Confidence:** 4

**Summary:**

This paper proposes OPPO (Omni-Perception Policy Optimization), a reinforcement learning framework for improving multimodal emotion reasoning in Omni-MLLMs. It introduces an Omni-Perception Reward that encourages models to cover fine-grained visual, audio, and emotion cues during reasoning, together with an Omni-Perception Loss that applies modality-specific KL regularization under unimodal masking to reduce cross-modal hallucinations. The authors also construct MEP-Bench, a diagnostic benchmark that measures cue utilization and perception faithfulness. Experiments on MER-UniBench show that OPPO improves both perception metrics and task performance, achieving state-of-the-art results compared with existing multimodal emotion reasoning baselines.

**Compliance With Llm Reviewing Policy:**

Affirmed.

**Final Justification:**

I maintain my original score: weak accept.

**Key Questions For Authors:**

See Weaknesses.

**Limitations:**

The paper includes an impact statement discussing potential applications. However, the limitations of the proposed method are not thoroughly discussed. In particular, it would be helpful to further discuss potential limitations related to the perception optimization pipeline and the construction of the proposed benchmark.

**Strengths And Weaknesses:**

Strengths:

1. The paper identifies two important issues in multimodal emotion reasoning-underutilization of multimodal cues and cross-modal hallucination.
2. The method is evaluated on MER-UniBench across multiple datasets and tasks.


Weaknesses:

1. The proposed framework mainly combines existing components, including reward-based evidence coverage and mask-based perception regularization mechanisms that encourage modality-grounded reasoning under masked inputs. Similar motivations have also been explored in perception-aware optimization approaches such as PAPO.
2. The approach relies on several manually designed components, including cue extraction using GPT, similarity thresholds, masking ratios, and reward weights. This design may limit robustness and generalization to other tasks or modalities.
3. The proposed MEP-Bench is derived from an existing dataset and relies on automatically extracted cues and embeddings, which may introduce noise or evaluation bias. The paper does not clearly describe all steps of the benchmark construction or analyze the reliability of these annotations, raising concerns about the reliability of the benchmark.
4. The cue extraction process relies on GPT-5, but the paper does not evaluate the reliability of the extracted cues or the sensitivity to the choice of the extraction model. Since the proposed reward directly depends on these cues, noise in the extraction pipeline may affect training stability and evaluation validity.
5. The cue–clause matching relies on Qwen3-Embedding similarity, but the paper does not evaluate the sensitivity to different embedding models. It would be helpful if the authors could clarify whether different embedding models affect the similarity distribution and the choice of the matching threshold.
6. The cue–clause matching assigns each clause to a single cue based on embedding similarity, but reasoning sentences may contain cues from multiple modalities within the same clause. For example, Table 8 includes sentences referencing “facial expression, body movements, tone, and subtitle content” simultaneously. It is unclear how such mixed-cue cases are handled, and this design may introduce noisy modality supervision.
7. There may be cases where the original emotion label becomes invalid after masking a modality (e.g., when the removed modality contains the primary emotional evidence). It would be helpful if the authors could clarify whether such situations were considered and how the method handles potential label mismatch under masked inputs.
8. The paper lacks failure case analysis, which would help better understand the limitations of the proposed method.

---

> ### Author Rebuttal · Authors · 2026-03-31
>
> ## To Reviewer wuw6
>
> We sincerely thank the reviewer for the detailed evaluation. We address your concerns below.
>
> **R1: Comparison with PAPO.** We contrast our approach with PAPO across three key dimensions:
> * **Training Stability:** PAPO’s unbounded KL on all tokens disrupts modality-agnostic distributions, causing collapse and requiring empirical entropy patches. OPPO restricts KL exclusively to modality-grounded tokens, ensuring stability without additional regularization.
> * **Resolving Unfaithfulness:** PAPO cannot reliably address cross-modal hallucination. Since its KL constraint applies to all tokens, models can satisfy it via global shifts, entangling modality-specific evidence with unrelated tokens. OPPO uses precise routing to maximize KL strictly on modality-specific tokens, ensuring statements are causally driven by actual multimodal signals.
> * **Unified Synergistic Framework:** The Omni-Perception Reward anchors factual evidence, preventing "cheating" where models inflate KL by simply degrading the masked distribution. This synergy locates genuine cues while eliminating optimization shortcuts (see in Appendix A).
>
> **R2: Component Robustness and Generalization.** Our ablation studies demonstrate that most hyperparameter combinations yield positive gains over the baseline; we simply report the optimal configuration.
>
> **R3: MEP-Bench Details.** We will expand benchmark details in the revised version. We filtered 322 human-annotated OV-MERD reasoning chains into 300 instances by removing samples missing single-modality cues, and all extracted cues were manually verified to eliminate automated bias. Additionally, 500 POPE-style probes were selected to target strongly correlated audio-visual cues to test hallucinations.
>
> **R4: Extraction Quality and Sensitivity.** Evidence extraction is a simple task where modern LLMs excel. Across GPT-5, Gemini-3.1, and DeepSeek-V3.1, human validation (200 instances) shows >94% accuracy (Table 1) and >0.92 pairwise semantic similarity (Table 2). Furthermore, retraining our model with Gemini-extracted rewards yields performance identical to the GPT-5 baseline (Table 3). These results demonstrate that OPPO is highly robust to cue extraction.
>
> **Table 1: Human Evaluation of Cue Extraction Accuracy**
> |Model|Visual %|Audio %|Emotion %|Avg %|
> |-|-|-|-|-|
> |GPT-5|93.6|95.5|96.9|95.3|
> |Gemini-3.1|92.3|94.4|96.6|94.4|
> |DeepSeek-V3.1|92.5|94.0|96.3|94.2|
>
> **Table 2: Overall Semantic Similarity of Extracted Cues**
> |Model Pair|Mean Similarity|
> |-|-|
> |Gemini vs. GPT-5|0.9234|
> |Gemini vs. DeepSeek|0.9230|
> |GPT-5 vs. DeepSeek|0.9280|
>
> **Table 3: Task Performance with Different Extraction Models**
> |Extraction Model|Senti.|Basic.|Fine.|Mean|
> |-|-|-|-|-|
> |GPT-5 Original|85.83|76.18|67.09|79.45|
> |Gemini-3.1|86.37|75.75|66.42|79.43|
>
> **R5: Sensitivity to Embedding Models.** We compared Qwen3-Embedding-0.6B, Qwen3-Embedding-4B, and BGE-M3. Table 4 shows that different models naturally yield different absolute scores. Table 5 tracks coverage hit rates across thresholds, which must be tailored to each model's distribution to retain valid signals while filtering low-quality matches. We conducted experiments using 0.5 as the threshold for the Qwen models and 0.6 for BGE-M3. As shown in Table 6, the final reasoning performance is highly comparable across all three embedding architectures.
>
> **Table 4: Average Top-1 Similarity Scores**
> |Metric|Qwen-0.6B|Qwen-4B|BGE-M3|
> |-|-|-|-|
> |Top-1 Mean|0.6869|0.6023|0.7121|
>
> **Table 5: Threshold Coverage Hit Rates**
> |Threshold|Qwen-0.6B|Qwen-4B|BGE-M3|
> |-|-|-|-|
> |0.50|0.914|0.933|0.997|
> |0.55|0.790|0.728|0.969|
> |0.60|0.654|0.479|0.872|
>
> **Table 6: Final Task Performance Across Embeddings**
> |Embedding Model|Senti.|Basic.|Fine.|Mean|
> |-|-|-|-|-|
> |Qwen-0.6B|86.40|79.18|67.16|81.05|
> |Qwen-4B|86.33|78.37|67.38|80.69|
> |BGE-M3|86.87|77.69|67.15|80.60|
>
> **R6: Handling Mixed-Cue Clauses.** Regarding the example in Table 8, our pipeline segments reasoning texts into fine-grained clauses via punctuation (including commas). Thus, sentences referencing multiple cues—facial expressions, body movements, tone, and subtitles—are automatically split into distinct segments. This ensures each clause isolates a single modality, preventing noisy supervision.
>
> **R7: Label Mismatch Under Masking.** To clarify, we compute separate unmasked and masked RL rollout paths. Masking exclusively serves the KL penalty to differentiate the two distributions. Crucially, final emotion evidence and rewards are always derived from the unmasked path; thus, masking never impacts training outputs and is entirely absent during inference.
>
> **R8: Failure Cases.** We recognize that our current framework does not explicitly account for missing modality scenarios such as off-screen voiceovers. In such cases, emotion analysis relies heavily on implicit cross-modal correlations. We will discuss these limitations and explore adaptive perception mechanisms in our future work.

---

> > ### Author Rebuttal · Reviewer_wuw6 · 2026-04-03
> >
> > I would like to thank the authors for their rebuttal. I have decided to maintain my score.

---

> > > ### Author Response · Authors · 2026-04-06
> > >
> > > We sincerely thank you for taking the time to review our rebuttal. We appreciate your constructive feedback throughout the review process and your continued support of our work.

---

### Official Review · Reviewer_gQJE · 2026-03-09

**Soundness:** 3
**Presentation:** 3
**Significance:** 2
**Originality:** 3
**Overall Recommendation:** 4
**Confidence:** 4

**Summary:**

This paper optimizes the RL process of the MER task. By constructing a diagnostic benchmark, it identifies that current methods do not sufficiently guarantee the reliability of perception during reasoning. Correspondingly, it proposes a dual-component solution to tackle this. Specifically, an Omni-Perception Reward is proposed to encourage coverage of salient cues, and an Omni-Perception Loss is introduced to distinguish the distribution of tokens with/without the necessary multimodal context. Comprehensive evaluation on MER-UniBench demonstrates the effectiveness of the proposed algorithm.

**Compliance With Llm Reviewing Policy:**

Affirmed.

**Final Justification:**

This paper is clearly presented and methodologically sound, making it a solid and qualified submission overall. However, the incremental nature of its technical contributions and the fairly standard experimental results prevent me from recommending a higher score.

**Key Questions For Authors:**

1. I am wondering about the benefits of first extracting evidence and then use embedding model to measure similarity compared to LLM-as-judge (both coarse-grained on the entire CoT and fine-grained on each extracted evidence).
2. Can the authors demonstrate why Omni-Perception Loss improves training stability?
3. Can the authors provide more details for the MEP-bench?

**Limitations:**

I think the primary limitation is the lack of emotion-oriented designs. The paper mainly handles the perception by referring to the ground-truth, and authors are encouraged to delve deeper into more flexible evaluation and the reasoning from factual cues to emotional states.

**Strengths And Weaknesses:**

Strengths:
1. The paper optimizes the reasoning process of multimodal LLMs, which is a popular and meaningful topic in the field. The motivation for enhancing perception reliability is reasonable and well-justified by the diagnostic datasets.
2. The method is introduced clearly and is easy to follow. The proposed component aligns well with the two perception perspectives, namely *utilization* and *faithfulness*. The provided implementation details and code offer sufficient reproducibility.
3. The experiments validate the effectiveness of the proposed method. The detailed ablation experiments further demonstrate the rationalities behind design choices.

Weakness:
1. While the paper's overall quality and presentation are good, I think the methodology contributions are somewhat incremental for ICML (possibly enough for AAAI/ACMMM).
a. From the perspective of multimodal reasoning studies, the Omni-Perception Reward can hardly distinguish itself from the LLM-as-judge rewards, since it inherently still relies on the semantic correlation measured by LLMs. Also, compared to the baseline PAPO, the Omni-Perception Loss only introduces variants on the masked component (in a highly intuitive way) and applied tokens (a widely-adopted technique, i.e., VPPO[1]).
b. From the perspective of MER studies, the proposed methods introduce limited emotion-oriented designs. Instead, the optimization objective of mimicking cues within the ground truth CoTs could potentially set constraints on models' exploration behaviour, and may not be beneficial for maintaining the subjectivity of emotion interpretation.
  c. In summary, I think a paper should contribute enough to at least one of the fields, and the paper currently has not met this requirement.
2. Besides the method, the proposed MEP-bench could have contributed to the significance of the paper. However, the description for it is too vague and is therefore hard to guarantee its quality and the guidance for future studies.
3. The experiment section spends too many spaces on less significant hyperparameter selection, instead of insightful analysis on the proposed method. In my opinion, some analysis on how the fine-grained Omni-Perception Reward/using the qwen3-embedding model outperforms LLM-as-judge, or how the token-level loss improves training stability (and some other analysis) can improve the depth and quality of the paper.

[1] Spotlight on Token Perception for Multimodal Reinforcement Learning. ICLR 2026.

---

> ### Author Rebuttal · Authors · 2026-03-31
>
> ## To Reviewer gQJE
> We sincerely thank the reviewer for recognizing the overall quality of our paper. We address your insightful concerns below.
>
> **R1: Embedding Similarity vs. LLM-as-Judge.** Expanding on Section 3.3, we detail three advantages of our embedding-based method:
> * **Synergistic Framework Design:** This is our core methodological contribution. To effectively suppress cross-modal hallucination, our Omni-Perception Loss requires precise token-level routing. An LLM judge only provides a detached scalar reward and cannot perform this token routing. Conversely, our method calculates a continuous reward while constructing the Evidence-Routing Matrix for the KL penalty. As a result, LLM-as-a-Judge cannot resolve the unfaithfulness issue, because it cannot distinguish whether a “better” reasoning chain is grounded in modality-specific cues or merely produced by hallucinating plausible-sounding explanations.
> * **Signal Stability:** Recent studies [1, 2] reveal that LLM evaluators suffer from rating indeterminacy and high variance. Embedding similarity provides a deterministic and continuous semantic gradient, avoiding the optimization instability inherent to LLM judges.
> * **Empirical Superiority and Efficiency:** We conducted ablations using two LLM-as-a-judge variants. Specifically, we utilized Qwen3-Instruct-14B as the evaluator model, deploying it on two H100 GPUs via vLLM. The coarse-grained judge scores the entire reasoning chain from 1 to 10. The fine-grained judge verifies each extracted cue individually. We trained model variants using only these rewards. As shown in Table 1, our reward outperforms both LLM judge variants in downstream performance. Furthermore, measuring the reward computation time per sample reveals our method is significantly faster while achieving superior performance.
>
> **Table 1: Task Performance and Efficiency**
> |Reward Type|Senti.|Basic.|Fine.|Mean|Time(s)|
> |-|-|-|-|-|-|
> |Coarse LLM Judge|86.17|71.95|67.58|77.75|0.15|
> |Fine-grained LLM Judge|84.97|74.63|66.14|78.28|0.35|
> |Omni-Perception Reward|85.83|76.18|67.09|79.45|0.07|
>
> **R2: Training Stability and Comparison with PAPO.** In the original PAPO paper, the authors observed performance collapse during RL training, attributing it to the fact that maximizing a KL divergence is theoretically unbounded. To maintain stability, they introduced an empirically inspired Double Entropy Regularization patch. When implementing a naive PAPO baseline, we indeed encountered similar performance collapses. We argue that applying the masked KL penalty to **all tokens** exacerbates this issue. It pushes modality-agnostic information away from its natural distribution, thereby making the optimization inherently fragile.
>
> Our approach directly alleviates this instability by strictly restricting the KL loss to the specific tokens. Furthermore, the Omni-Perception Reward acts as a structural anchor that guides the model to output factual evidence for each modality. This prevents a cheating behavior where the model artificially inflates the KL divergence simply by degrading the masked distribution rather than improving the genuinely grounded distribution. Extensive experiments in the paper confirm that our framework remains stable throughout training, even without the need for Double Entropy Regularization.
>
> **R3: Emotion-Oriented Design.** We respectfully clarify that our motivation stems directly from a critical bottleneck in current multimodal emotion reasoning tasks. MLLMs lack reliable perception in omni emotion reasoning, causing severe evidence underutilization and unfaithfulness. OPPO directly resolves this flaw. The Omni-Perception Reward grounds reasoning on genuine multimodal evidence, securing a reliable factual foundation for subjective interpretation, as validated by our ablation studies.
>
> **R4: MEP-Bench Construction Details.**  We will include a comprehensive appendix detailing MEP-Bench in the revised version. Key construction details:
> * **Data Source:** We built MEP-Bench upon the OV-MERD[3], which provides the highest quality human-annotated reasoning rationales. We filtered out samples missing single-modality cues, retaining 300 robust and cue-rich instances.
> * **Utilization:** To ensure reproducibility, we utilized the open-source Qwen2.5-7B-Instruct to automatically extract cues for utilization evaluation. We manually verified these extracted cues to ensure they accurately reflect the ground truth.
> * **Faithfulness:** We specifically targeted cues with strong audio-visual correlations, as illustrated in Figure 1b. We then transformed these cues into POPE-style binary questions. In total, we generated 500 probing questions targeting the audio and visual modalities to test cross-modal hallucinations under masked conditions.
>
> [1] Validating LLM-as-a-judge systems under rating indeterminacy. [2] Examining reasoning LLM-as-judges in non-verifiable LLM post training. [3] Explainable Multimodal Emotion Recognition.

---

> > ### Author Rebuttal · Reviewer_gQJE · 2026-04-02
> >
> > Thank you for your careful and detailed rebuttal. I acknowledge the methodological effectiveness of the proposed design in your manuscript. However, my primary concern remains the somewhat incremental nature of the overall contribution.
> >
> > 1. Regarding the Omni-Perception Reward and RL Stability (R1 & R2): While the Omni-Perception reward indeed outperforms the LLM-as-a-judge baselines, it essentially functions by converting the latter from an online evaluation into an offline version while increasing discriminability (an effect that could arguably be achieved through proper normalization techniques). I do appreciate your argument that token-level weighting helps alleviate training instability. Nevertheless, the approach still overlaps considerably with existing RL literature, though I understand that finding completely orthogonal directions is challenging in such a saturated research field.
> > 2. Regarding the Emotion-Oriented Design (R3): I respectfully disagree with your rationale on this point. Perception is fundamentally indispensable for any multimodal task, meaning your justification could easily be transferred to almost any other similar multimodal reasoning task. It does not feel uniquely necessary for emotion-oriented tasks.
> >
> > Personally, I do not believe a paper must introduce inspiring innovations in both directions simultaneously. However, the aggregate contribution across these two aspects is still not entirely satisfying to me. Despite my reservations, taking into consideration the positive feedback from the other reviewers and the average quality of papers in my reviewing batch, I am willing to raise my score to a 4. I hope the authors will incorporate more details regarding the dataset and include some more insightful experiments.

---

> > > ### Author Response · Authors · 2026-04-06
> > >
> > > Dear Reviewer,
> > >
> > > We sincerely thank you for your professional attitude and the constructive discussion. We are very glad that our previous rebuttal resolved some of your initial concerns.
> > >
> > > To address your remaining questions regarding the overall contribution of our work, specifically our RL design compared to existing literature (such as VPPO[1]) and the necessity of our holistic OPPO framework for omni-modal reasoning, we have conducted additional experiments and comparisons. We hope this further alleviates your concerns.
> > >
> > > ### 1. Comparison with VPPO
> > > To highlight our methodological distinction, we compare OPPO with the existing RL literature you mentioned, i.e., VPPO.
> > >
> > > The methodological differences between VPPO and OPPO are as follows:
> > > * **VPPO:** First masks partial images to compute KL divergence, identifying reasoning trajectories and visual tokens with high dependency. It then weights these trajectories and applies a GRPO loss specifically to the high-dependency tokens.
> > > * **OPPO:** First identifies crucial tokens at the **semantic level**, and then uses KL divergence as a constraint to optimize them.
> > >
> > > We point out that VPPO's design inherently cannot resolve the hallucination problem caused by spurious audio-visual correlations in an omni-modal setting.
> > > For instance, due to spurious correlations, a model might falsely associate an angry audio tone with a visual "frown" (even if the person's face is neutral). Under VPPO's mechanism, if the audio is masked, the hallucinated "frown" token will disappear (since it is generated by the audio hallucination), exhibiting high dependency. VPPO will then incorrectly identify this hallucinated word as a key token and optimize the policy toward it, exacerbating the hallucination.
> > >
> > > We implemented VPPO in our omni-modal setting. As shown in the table below, VPPO underperforms our baseline. This demonstrates that OPPO's "semantics-first, KL-constraint" design is structurally necessary for omni-modal reasoning.
> > >
> > > **Table A: Performance of VPPO in the Omni-modal Setting**
> > >
> > > | Method | OV-MERD+ | MER23 | MER24 | MELD | IEMOCAP | MOSI | MOSEI | SIMS | SIMSv2 | Mean |
> > > | :--- | :--- | :--- | :--- | :--- | :--- | :--- | :--- | :--- | :--- | :--- |
> > > | **VPPO** | 66.35 | 79.39 | 82.45 | 58.32 | 63.22 | 84.69 | 85.56 | 85.94 | 86.51 | **76.93** |
> > > | **OPPO** | 67.16 | 87.73 | 90.34 | 64.06 | 73.60 | 86.50 | 84.63 | 86.22 | 88.26 | **81.05** |
> > > ---
> > >
> > > ### 2. Effectiveness of the Holistic OPPO Framework
> > > OPPO is designed as an inseparable framework where its two core components, the Omni-Perception Reward and the Omni-Perception Loss, dynamically support each other. To demonstrate this synergy, we extended our ablation studies by removing the Omni-Perception Reward and using only the Evidence-Routing Matrix to guide the Omni-Perception Loss.
> > >
> > > **Results:** Without the Omni-Perception Reward explicitly pushing the model to utilize key evidence cues, the critical tokens identified by the Evidence-Routing Matrix are suboptimal. Consequently, applying the Omni-Perception Loss in isolation yields limited improvements.
> > >
> > > **Table B: Ablation using only Omni-Perception Loss**
> > >
> > > | Method | OV-MERD+ | MER23 | MER24 | MELD | IEMOCAP | MOSI | MOSEI | SIMS | SIMSv2 | Mean |
> > > | :--- | :--- | :--- | :--- | :--- | :--- | :--- | :--- | :--- | :--- | :--- |
> > > | **Loss Only** | 66.92 | 82.60 | 85.43 | 61.11 | 68.65 | 84.54 | 85.98 | 85.50 | 86.73 | **78.60** |
> > > | **OPPO** | 67.16 | 87.73 | 90.34 | 64.06 | 73.60 | 86.50 | 84.63 | 86.22 | 88.26 | **81.05** |
> > >
> > > ***
> > >
> > > **We deeply appreciate your recognition of the overall quality of our paper. Per your suggestion, we will incorporate these experiments and a more detailed introduction to the datasets into the revised appendix. Thank you again for your time and constructive feedback!**
> > >
> > > [1] Spotlight on Token Perception for Multimodal Reinforcement Learning. ICLR 2026.

---

### Official Review · Reviewer_UGqP · 2026-03-10

**Soundness:** 3
**Presentation:** 3
**Significance:** 3
**Originality:** 3
**Overall Recommendation:** 4
**Confidence:** 3

**Summary:**

This paper studies Multimodal Emotion Reasoning (MER) with omni-modal multimodal large language models (Omni-MLLMs), where models are expected to predict emotions and generate explanations grounded in visual, acoustic, and textual signals. The authors find that existing models often underutilize multimodal cues and may produce unfaithful reasoning by hallucinating modality-specific evidence. To address these issues, they propose OPPO (Omni-Perception Policy Optimization), a reinforcement learning framework that improves multimodal perception through an Omni-Perception Reward, which encourages recovery of fine-grained modality cues, and an Omni-Perception Loss, which penalizes cross-modal hallucinations by comparing full and modality-masked inputs. The paper also introduces MEP-Bench, a diagnostic benchmark for evaluating multimodal cue utilization and reasoning faithfulness. Experimental results show that OPPO achieves state-of-the-art performance on MER-UniBench and improves both utilization and faithfulness metrics, highlighting the importance of optimizing omni-modal perception for reliable multimodal emotion reasoning.

**Compliance With Llm Reviewing Policy:**

Affirmed.

**Key Questions For Authors:**

1. What is the accuracy of the multimodal cues automatically extracted by GPT-5? Was any human validation conducted? If noise exists, how robust is OPPO to it?
2. Is it possible that the model "cheats" by generating text that is semantically similar to the cue embeddings, without truly relying on the corresponding visual or acoustic information?
3. Is OPPO applicable to other multimodal reasoning tasks, such as video QA, general multimodal reasoning, and robotics perception reasoning, or is it specific to emotion reasoning?

**Limitations:**

The authors do not explicitly discuss the limitations or potential negative societal impacts of their work in the paper. It might be helpful to also briefly discuss some potential considerations, such as the model's reliance on high-quality annotations, computational costs, generalization to diverse cultural contexts, or broader ethical implications of emotion recognition technologies.

**Strengths And Weaknesses:**

Strengths

1. The paper clearly identifies two core problems in current multimodal emotion reasoning (MER) models: insufficient utilization of modality evidence and cross-modal hallucination. Re-examining the MER task from the perspective of perceptual reliability provides a thought-provoking and practically relevant framing of the problem.
2. The two modules in OPPO are specifically designed to address these issues: the Reward module improves evidence utilization, while the Loss module enhances faithfulness. By applying token-level modality constraints, the method avoids the optimization instability caused by applying KL regularization to all tokens, as seen in approaches like PAPO. The overall method design is logically coherent and well-motivated.
3. The introduction of MEP-Bench provides a benchmark specifically designed to evaluate multimodal evidence recall and inference accuracy under modality masking. Compared with relying solely on final task metrics, this diagnostic benchmark is highly valuable for assessing the quality and reliability of multimodal reasoning.
4. The experimental evaluation is thorough and well-structured, including MER-UniBench multi-task assessment, perception diagnostic metrics, multiple baseline comparisons, and ablation studies. This comprehensive experimental setup ensures that the proposed method is rigorously validated from multiple perspectives.

Weaknesses

1. The paper uses GPT-5 to automatically extract evidence from reasoning texts but lacks discussion on extraction quality, annotation consistency, and the impact of noise.
2. The evidence coverage method relies on embedding similarity with a fixed threshold, which risks semantic mismatches and may encourage the model to generate superficially similar explanations rather than genuinely grounding its reasoning in visual or acoustic information.

---

> ### Author Rebuttal · Authors · 2026-03-31
>
> ## To Reviewer UGqP
>
> We sincerely thank the reviewer for the constructive feedback. Below, we address your concerns with detailed empirical evidence.
>
> **R1: Quality and Robustness of Evidence Extraction.** Extracting evidence (visual, audio, and emotion cues) from reasoning texts is a simple task without complex reasoning. Modern LLMs are highly capable of this. To validate this, we compared GPT-5, Gemini-3.1-Flash-Lite-Preview, and DeepSeek-V3.1 from three perspectives:
>
> * **Human Validation:** We randomly sampled 200 instances and manually scored the extraction accuracy for visual, audio, and emotion cues, penalizing for missing or hallucinated information. As shown in Table 1, all models achieve over 94% average accuracy, with minimal variance between them.
> * **Semantic Similarity:** We computed the pairwise cosine similarities of the extracted cues across these three models using Qwen-Embedding. As detailed in Table 2, the mean similarities across all modalities consistently exceed 0.9, proving strong semantic alignment.
> * **Training Robustness:** To isolate the impact of different LLMs, we trained a model variant using only the Omni-Perception Reward guided by Gemini-extracted cues. The final performance shown in Table 3 is nearly identical to the GPT-5 counterpart, demonstrating that our reward mechanism is highly robust to the choice of extraction models and resilient to minor extraction noise.
>
> **Table 1: Human Evaluation of Cue Extraction Accuracy**
>
> | Model | Visual % | Audio % | Emotion % | Avg % |
> | :--- | :--- | :--- | :--- | :--- |
> | GPT-5 | 93.6 | 95.5 | 96.9 | 95.3 |
> | Gemini-3.1 | 92.3 | 94.4 | 96.6 | 94.4 |
> | DeepSeek-V3.1 | 92.5 | 94.0 | 96.3 | 94.2 |
>
> **Table 2: Pairwise Semantic Similarity of Extracted Cues**
>
> | Model Pair | Visual | Audio | Emotion | Mean |
> | :--- | :--- | :--- | :--- | :--- |
> | Gemini vs. GPT-5 | 0.9137 | 0.9018 | 0.9547 | 0.9234 |
> | Gemini vs. DeepSeek | 0.9169 | 0.9004 | 0.9518 | 0.9230 |
> | GPT-5 vs. DeepSeek | 0.9170 | 0.9145 | 0.9526 | 0.9280 |
>
> **Table 3: Task Performance with Different Extraction Models**
>
> | Extraction Model | Senti. | Basic. | Fine. | Mean |
> | :--- | :--- | :--- | :--- | :--- |
> | GPT-5 Original | 85.83 | 76.18 | 67.09 | 79.45 |
> | Gemini-3.1 | 86.37 | 75.75 | 66.42 | 79.43 |
>
> **R2: Prevention of Cheating via Semantic Guesses.** We address the concern that the model might cheat by generating text semantically similar to the cue embeddings without genuinely relying on the corresponding visual or acoustic information. We highlight that **our Omni-Perception Loss is specifically designed to eliminate this cheating behavior**.
>
> During training, we evaluate the Kullback-Leibler divergence between the token output distribution under the **full multimodal input** and **a counterfactual input** where a specific modality is masked. Crucially, we apply this divergence constraint exclusively to the generated tokens that semantically map to that specific modality.
>
> If the model attempts to cheat by blindly guessing visual or acoustic features to maximize Omni-Perception Reward without actually attending to the respective sensory signals, its token output distribution would remain largely unchanged when that target modality is masked. By explicitly maximizing this divergence on modality-specific evidence tokens, **our framework heavily penalizes any ungrounded guessing**. This synergistic design guarantees that even if the model miraculously guesses the correct visual or acoustic description, it will be penalized unless that generation is causally driven by the true multimodal input.
>
> **R3: Applicability to General Tasks.** Yes, OPPO is applicable to broader multimodal reasoning tasks. We initially focused on emotion reasoning because we identified perception reliability as a critical bottleneck specifically within this domain, and the corresponding multimodal evidence is easily extractable. Our framework was designed to resolve the severe underutilization and unfaithfulness inherent to emotion reasoning. However, this core methodology can also be applied to other omni-modal domains facing similar perception bottlenecks. For tasks requiring outputs to be anchored on modality-grounded evidence, our framework provides a symmetrical solution: the Omni-Perception Reward actively guides the discovery of genuine multimodal cues, while the masked KL penalty regularizes against ungrounded hallucinations.

---

> > ### Author Rebuttal · Reviewer_UGqP · 2026-04-03
> >
> > I would like to thank the authors for their rebuttal. I have decided to maintain my score.

---

> > > ### Author Response · Authors · 2026-04-06
> > >
> > > Thank you for reviewing our rebuttal and for your continued support of our work.

---

### Decision · Program_Chairs · 2026-04-30

**Decision:**

Accept (regular)

**Comment:**

The paper originally received scores of weak accept and 1 weak reject. After the rebuttal, the weak reject score raised to weak accept. Overall, the reviewers are positive on this paper.

Strengths of the paper include important and clear problems are addressed, solid evaluations across multiple datasets and tasks, clear description of proposed method, potential to have influence beyond emotion recognition.

Weaknesses of the paper include lack of discussion on AI extraction quality, some missing experiments (shown in rebuttal) and missing details on sensitivity to embedding models.

Overall, the strengths outweigh the weaknesses.